# Molecular organization of fibroin heavy chain and mechanism of fibre formation in *Bombyx mori*

**Rafael O. Moreno-Tortolero** [ORCID][1,2] ✉**, Yijie Luo** [ORCID][1,3]**, Fabio Parmeggiani**[1,3,4]**, Nick Skaer**[5]**, Robert Walker**[5]**,
**Louise C. Serpell** [ORCID][6]**, Chris Holland**[7] **& Sean A. Davis** [ORCID][1] ✉

Fibroins' transition from liquid to solid is fundamental to spinning and underpins the impressive native properties of silk. Herein, we establish a fibroin heavy chain fold for the Silk-I polymorph, which could be relevant for other similar proteins, and explains mechanistically the liquid-to-solid transition of this silk, driven by pH reduction and flow stress. Combining spectroscopy and modelling we propose that the liquid Silk-I fibroin heavy chain (FibH) from the silkworm, *Bombyx mori*, adopts a newly reported β-solenoid structure. Similarly, using rheology we propose that FibH N-terminal domain (NTD) templates reversible higher-order oligomerization driven by pH reduction. Our integrated approach bridges the gap in understanding FibH structure and provides insight into the spatial and temporal hierarchical self-assembly across length scales. Our findings elucidate the complex rheological behaviour of Silk-I, solutions and gels, and the observed liquid crystalline textures within the silk gland. We also find that the NTD undergoes hydrolysis during standard regeneration, explaining key differences between native and regenerated silk feedstocks. In general, in this study we emphasize the unique characteristics of native and native-like silks, offering a fresh perspective on our fundamental understanding of silk-fibre production and applications.

The major protein component of silk, fibroin, is stored in arthropods as a liquid gel before undergoing a triggered conversion via spinning, into a highly stable solid fibre. This transformation starts within the silk gland, a highly specialised sack-shaped organ where not only the silk proteins are produced but where tight control over pH and metal ion concentrations is exerted[1–5]. However, there is a gap in our knowledge surrounding the molecular level events during this transformation. There are some commonly agreed principles, including the contribution of pH and flow stress/strain, energy input/work[6], in triggering assembly and promoting structural conformational changes[7,8]. Though, there is less consensus regarding protein conformational changes. In particular, the molecular structure of the liquid state silk, often called silk-I, is heavily debated[9–13] and is the focus of our research.

For silks' model organism *Bombyx mori*, fibroin, both within the gland and in the fibre, is composed of three different proteins, fibroin heavy chain (FibH), fibroin light chain (FibL) and a glycoprotein, fibrohexamerin (P25)

of molecular masses 392, 28 and 25 KDa, respectively. These form a complex known as the elementary unit with molar ratios of 6:6:1, wherein FibH and FibL form a heterodimer stabilized by a single disulphide linkage located at the C-terminal domain of FibH and the interaction between these and P25 is non-covalent[14,15]. However, the structural details of this system are not well known, save for the crystal structure of the N-terminal domain (NTD) of FibH. It is generally accepted that only FibH is essential for fibre formation, with FibL and P25 being auxiliary in the secretion process[16–18], with some silk-producing moths and other related species lacking these latter proteins entirely[19–21]. Accordingly, due to its dominant mass contribution, FibH is thought to be the main protein responsible for the properties of fibroin across different length-scales, from the NMR chemical shifts[22], to the diffraction patterns of Silk-I (liquid state) and Silk-II (fibre)[23].

Silk-I has previously been termed α-silk, as it was erroneously assumed to include α-helical folds, to distinguish between this conformational polymorph and the better characterised Silk-II, which is known to be rich in

[1]School of Chemistry, University of Bristol, Cantock's Close, Bristol BS8 1TS, UK. [2]Max Planck-Bristol Centre for Minimal Biology, University of Bristol, Bristol BS8 1TS, UK. [3]School of Biochemistry, University of Bristol, University Walk, Bristol BS8 1TD, UK. [4]School of Pharmacy and Pharmaceutical Sciences, Cardiff University, Redwood Building, King Edward VII Ave, Cardiff CF10 3NB, UK. [5]Orthox Ltd, Milton Park, 66 Innovation Drive, Abingdon OX14 4RQ, UK. [6]Sussex Neuroscience, School of Life Sciences, University of Sussex, Brighton BN1 9QG, UK. [7]Department of Materials Science and Engineering, University of Sheffield, Mappin Street, Sheffield S1 3JD, UK. ✉e-mail: ro.morenotortolero@bristol.ac.uk; s.a.davis@bristol.ac.uk

antiparallel β-sheet structures[24]. The exact structure of silk-I remains ambiguous, and it is often classified as either a random coil or an intrinsically disordered state[25,26]. However, discrete reflections in X-ray diffraction data suggest neither of these classifications is fully adequate[14,15,27]. Over time, several models have been proposed to account for the diffraction patterns (summarised in Table S1), however, all of these suffer from limitations inherent to the technical difficulties in acquiring aligned fibre X-ray diffraction patterns or complete datasets from electron diffraction. Most of the proposed unit cells so far belong to low-symmetry systems, perhaps due to the initial required assumptions necessary to arrive at structural models given the limited data available. Currently, the most accepted Silk I model is a type-II β-turn rich structure[28], similar to the crankshaft model proposed by others before[29,30]. However, this model requires extensive intermolecular hydrogen bonding networks to persist[31], failing to account for the observation of similar chemical shifts both in dilute solutions and in the solid state measured by NMR, as well as small fibrillar structures observed herein and previously by electron microscopy[32].

Similarly, modelling of consensus sequences has indicated a right-handed β-helix showing lower relative free-energy values than the corresponding type-II β-turn[31]. Moreover, the observation of a range of distinct liquid crystalline textures (patterns) in vivo at the start of anterior section of the gland and the spinneret remains unexplained[33], with the mesogenic structures being unidentified. At the start of the anterior section of the silk gland, a "cellular optical texture" is observed, which transforms to an isotropic phase prior to reverting to a fully nematic phase before the spinneret[34]. The emergence of the cellular optical texture has been attributed to epitaxial anchoring of rod-like mesogens under confinement[35]. Yet, this model does not adequately account for the subsequent transition to a nematic texture, under flow, as the tube diameter in the gland decreases towards the spinneret. Curiously, at a similar position to the cellular optical texture, evidence of cholesteric order has been observed using electron microscopy[36]. More importantly, despite the evidence of supramolecular order, a transition from the Silk I to Silk II polymorph only occurs later near the spinneret[34,36].

In this study, we propose that at the molecular level, Silk-I corresponds to a folded fibrillar conformation of FibH. Furthermore, we rationalise how physicochemical triggers such as pH, metal ions, and stress/strain induce

conformational changes in this mesogen that result in concomitant supramolecular assembly or reorganization consistent with the observed optical textures inside the silk gland. The abrupt loss of the cellular optical texture derives from disruption of the preceding cholesteric phase, and is a key intermediate to facilitate the minimum energy transition to the ordered nematic phase, as previously proposed[34,36]. To investigate this hypothesis, we employed a comprehensive approach combining both computational simulations and experimental methods, encompassing various length scales ranging from the molecular to macroscopic. Our methodology involved the utilization of a fibroin solution achieved by dissolving silk fibres, after a proprietary degumming method, in concentrated LiBr solutions, resulting in a material referred to as native-like silk fibroin (NLSF). This NLSF exhibits numerous characteristics consistent with those of the native silk fibroin (NSF), including electrophoretic mobility and rheological properties. It is important to differentiate NLSF from standard regenerated silk fibroin (RSF)[37], as the latter fails to accurately replicate the molecular integrity and rheological properties of the native system.

## Results and discussion
### Molecular model creation and validation
Dilute solutions of NLSF were cast and left to dry slowly prior to being characterised by fibre X-ray diffraction (fXRD) experiments. When the incidence of the X-ray beam was perpendicular to the film, we obtained a pattern similar to those obtained by powder X-ray diffraction (pXRD), yet, as we rotated the film, sharper reflections were observed and eventually a highly oriented pattern emerged (Fig. 1a and Fig. S1), consistent with previous Silk-I data[30,38], see Table S2 for reflections. This data coupled with the observation of liquid crystalline phases suggest the presence of rod-like molecules, which would be preferentially oriented parallel to the surface of the film[39]. After running Alphafold2 (AF2) simulation of the repetitive domains of FibH[40–42], we found that these repetitive domains are predicted to adopt a novel β-solenoid conformation (Fig. S2). Briefly, the fold results from a supersecondary folding of strands, stabilised by inter-strand hydrogen bonds running along the solenoid axis, as depicted in Fig. 1b; the basic strand motif might explain the proposal of packing of almost extended strands by others for this polymorph[38,43–45]. Although in this case

**Fig. 1 | Structural model for fibroin heavy chain.**
**a** Experimental and simulated Silk-I diffraction pattern obtained from drop-cast films of NLSF and derived unit cell showing the proposed fibre axis. **b** Proposed unit cell comprised of AGXG motifs. **c** β-solenoid molecular model obtained for fibroin heavy chain, observed from the top and side views (top and bottom, respectively).

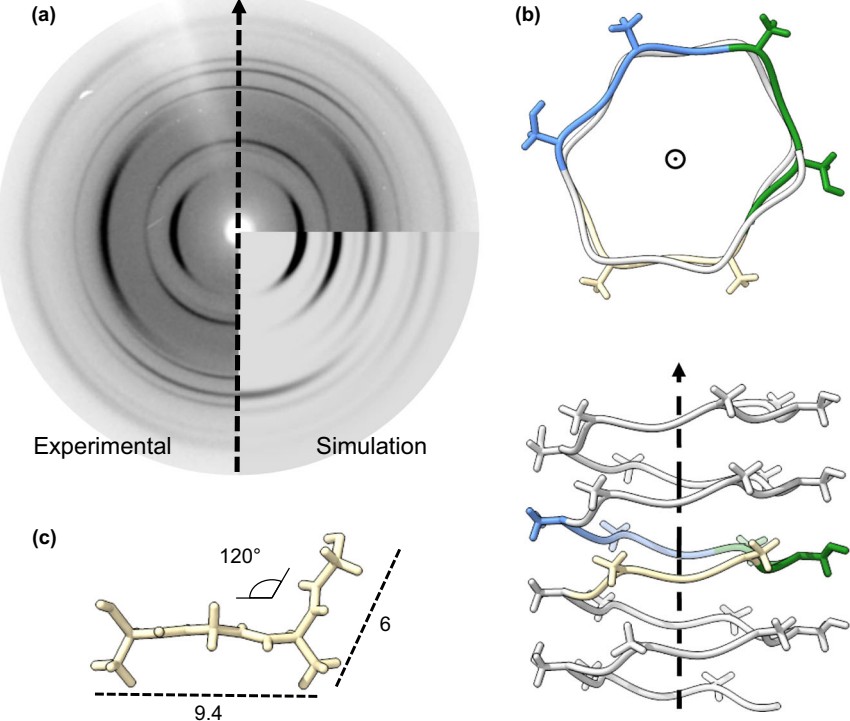

the abundance of glycine and the absence of a tightly packed core lead to motifs structurally closer to a polyproline-II or polyglycine-II configuration, which can be frequently assigned as disordered by CD spectroscopy[46]. It is possible that other G-rich proteins, such as fibroin heavy chain-like, spidroins, and others with hexapeptide repeated motifs of GX fold in similar structures[47]. Structural evidence of different spidroins folding into flexible worm-like structures was recently found by SAXS[48], pointing at the possibility that β-solenoids might be widespread among silk proteins. Comparable conformation and interpretations were found for the G-rich snow flea antifreeze protein, which was crystallized using racemates[49,50]. Similar β-helical configurations have also been suggested before for FibH, albeit mostly via computational approaches[12,31].

Although the predictions showed low confidence, and relatively high predicted Local Distance Difference Test (pLDDT, average ~50), all generated models maintained the solenoidal topology (Fig. S3) and were consistent with the topologies we observed by TEM (Fig. S4). Based on these results, we propose a trigonal unit cell containing a single curved strand, as depicted in Fig. 1c; initial unit cell parameters are a: 9.4, b: 3.4 and c: 6 Å, and α = β: ~90° and γ: 120°, although further refinement is required. We note that the proposed unit cell is similar to that proposed for the PolyGly-II polymorph[51], known to resemble Silk-I[28] and gives a simulated diffraction pattern that recapitulated the reflections obtained experimentally (Fig. 1a), allowing us to index most of the observed reflections as seen in Table S3. Nevertheless, observed discrepancies can be attributed to experimental constraints in aligning the sample, inherent heterogeneities in a real sample, and the software's inability to account for helical symmetry. This unit cell is

shown in context of the proposed solenoidal model in Fig. 1b. Given the markedly repetitive characteristics of FibH, featuring extensive low-complexity domains, characterized by a hexapeptide sequence (GAGAGX), where X may assume any of the residues A, S, Y, or V in descending order of abundance, twelve solenoidal domains are expected to form. The projected architecture of the protein consists of 12 solenoid bodies, interconnected by short linker regions that appear structurally unorganized, as depicted in Fig. S5. Significantly, within each repetitive domain, there are more regular and seemingly rigid subdomains (as shown in Fig. S3) that might be the ones giving rise to the deconvolved solid-state NMR Silk I* structures, with the remainder of the dynamic and unordered subdomains contributing to the overall Silk I spectrum[52,53].

Furthermore, both 1D proton (Fig. S6) and 2D ¹H-¹H TOCSY (Fig. S7), ¹H-¹³C HSQC, ¹H-¹⁵N HSQC (Fig. S8), ¹H-¹⁵N TROSY (Fig. 2a) and ¹H-¹H NOESY (Fig. 2a, b) were acquired. Assigned chemical shifts (Table S4) were very close to those reported elsewhere[22,54], as well as predicted shifts from our solenoid models (Table S5). Notably, we did not observe changes in the chemical shifts at pH 8 or pH 6, intended to replicate physiological pH change within the silk gland (Fig. S9). The assignment of the amide protons and nitrogen chemical shifts is depicted in Fig. 2a for simplified motifs within FibH. In addition, we note that when predicting the torsion angles for reduced motifs in FibH, two populations for G residues are possible within the generic GX motif[22]. One of these (ϕ, φ = 77°, 10°) was used to derive the type-II β-turn model for Silk-I*. However, the second population of torsion angles (ca. ϕ, φ = −80°, 180°) was not used to constrain the modelling and matches closely those found in the solenoidal model predicted by AF2 as

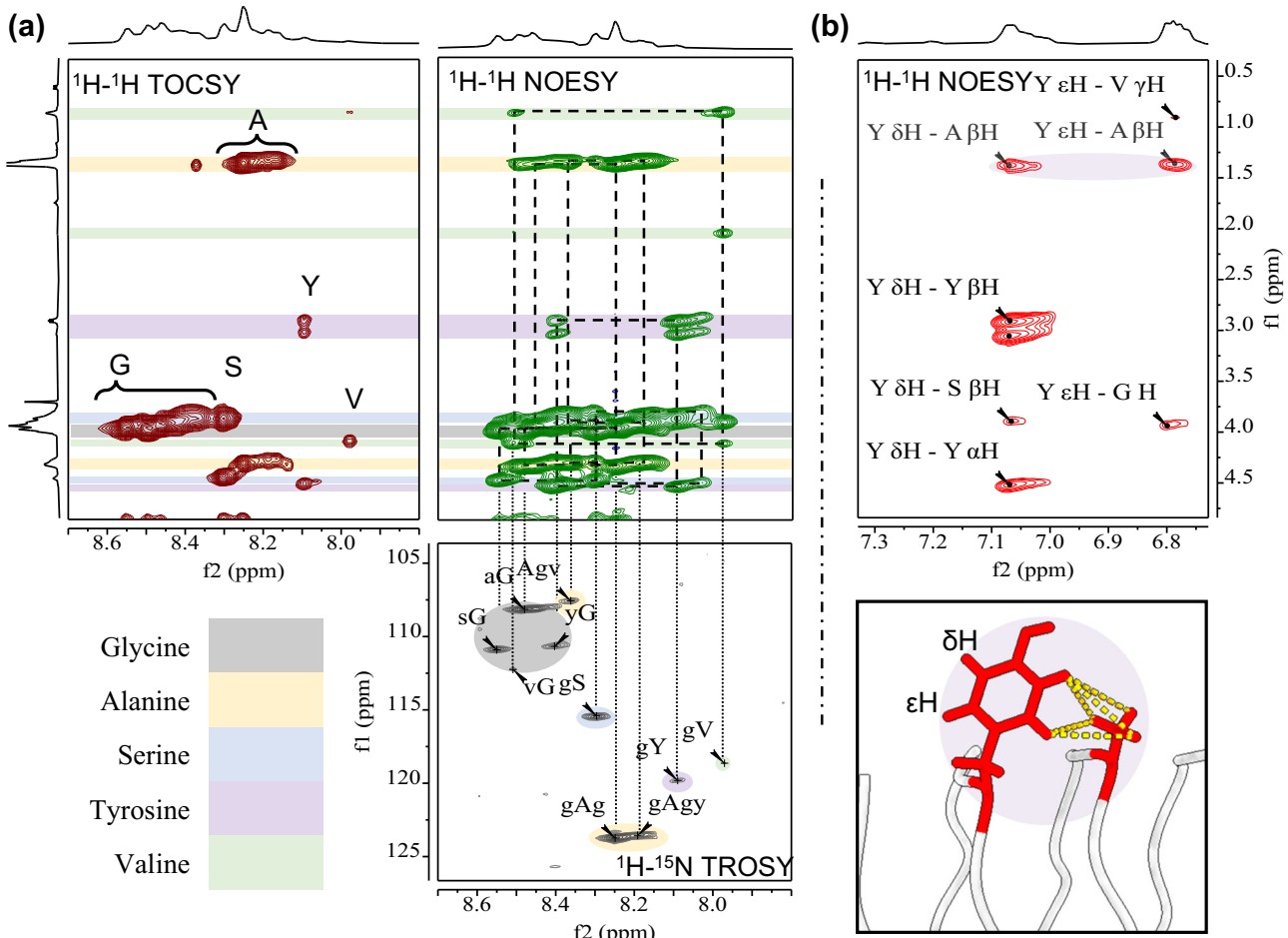

**Fig. 2 | NMR analysis of NLSF. a** Amide shift assignment via ¹H,¹H TOCSY, Amide region ¹H,¹H NOESY analysis, and ¹⁵N chemical shift assignment of simplified motifs via ¹H, ¹⁵N TROSY. **b** Analysis of the Tyrosine (Y) residue NOE signals

showing evidence of proximity to Alanie-CH, and cartoon representation of motif found within the Alphafold2 model showing positions like the measured via NOESY experiment.

**Fig. 3 | The effect of pH on rheological properties of NLSF, and the lost domains during regeneration. a** Master curves created using oscillatory shear data of the liquid and solid-like samples of NLSF; guide lines were added to aid in trend visualisation. **b** Cartoon showing the multidomain architecture of the primary structure of FibH (top), followed by an aligned cartoon representing the total sequence coverage from the hydrolysate peptides recovered from RSF. Curves showed averages (continuous lines) and standard deviation (shadowed area around average), technical replicates $N = 5$.

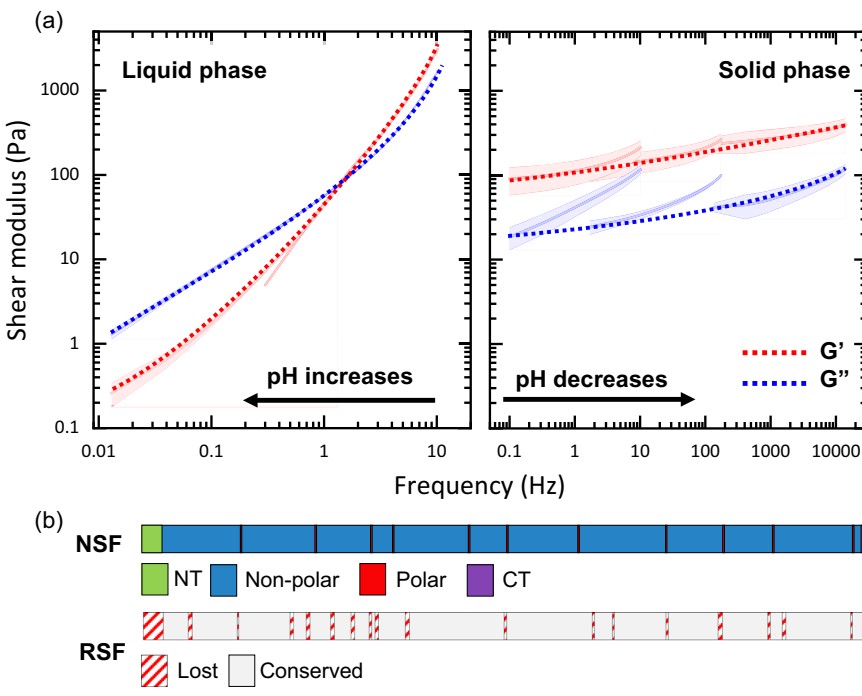

well as previously proposed angles from density functional theory (DFT) calculations on NMR values[13,55]. This conformation implies that in the transformation from Silk-I to Silk-II, G residues would not need to go through sterically hindered angles (see Fig. S10)[56], thus reducing the overall energetic cost of the Silk-I to Silk-II transformation. Further to these observations, NOESY experiments, shown in Fig. 2b evidences Y residues close to (~3.3 to 3.4 Å) the methyl sidechain of A, (Fig. S11), suggesting a possible methyl-pi interaction[57], which could be an inter-strand (within the solenoid) stabilising interaction. There are numerous Y residues on the predicted models, with the aromatic group hovering over methyl groups from A residues on contiguous strands (see Fig. 2b, bottom). Similarly, we observed other NOE intensities, at the amide proton from S residues, which although showing the strongest correlation with G, it also showed correlation with A-βH. Such patterns can be readily explained by our model, where the strongest correlation corresponds to abundant motifs gS (ca. 2.2 Å), and the weaker correlation with A-βH corresponding to across strand gS--A distance (2.7 Å), both the estimated distances and ratios are in good agreement with the experimental data (more in Supplementary Note: NOESY).

## The role of NTD, pH effect on rheological properties

It has largely been recognised that in native spinning both flow stress and pH changes are responsible for the transformation of the soluble fibroin into the solid fibre[7,58–60]. Rheological characterisation of native feedstock has provided snapshots into the strain-induced assembly. In most studies the silk dope is extracted from the posterior segment of the middle gland, where the pH is near neutral[61,62]. However, direct evidence of the structural changes induced by pH and their effects on rheological properties has proved elusive.

Here we used NLSF samples obtained at concentrations of 60 to 80 mg mL$^{-1}$, lower than those found inside the gland (190–300 mg mL$^{-1}$)[62], buffered at pH values to replicate the pH gradient within the silk gland, going from pH 8 in the posterior section down to 6.2 in the anterior section[63]. We observed that the material goes through a reversible sol-to-gel transition when buffered above pH 7, with the sample at pH 7 separating into two clear phases, gel-like and liquid-like. Single phases were observed for samples either below or above pH 7. Using NLSF at pH 7, under oscillatory rheology testing we were able to replicate many previous features seen in NSF (i.e., observation of crossover), with the exception of a lower overall modulus and viscosity and a lack of a single relaxation mode in the terminal region, both of which we attribute to a lower concentration and increased polydispersity of the NLSF against NSF studies[62]. Furthermore, we were able to uncover a reversible pH driven sol-gel transition occurring at a critical pH of 7, similar to the pH-dependent oligomerisation of NTD[64,65]. Figure 3a shows two master curves demonstrating the effect of pH on the obtained material; master curves were created from the data shown in Fig. S12 and the applied shifts are presented in Table S6; the linear viscoelastic limit (LVE) was determined beforehand, and plots are shown in Fig. S13. Decreasing the pH further turns the gel opaque irreversibly, providing evidence of larger aggregates forming. However, no evidence suggests that the pH drops below 6 within the gland.

For samples below pH 7, the observed shear modulus showed a reciprocal relationship with pH, increasing as the pH decreased. This suggests a more stable molecular network is formed as the pH decreases, evidenced by increased elastic modulus. Crucially, NLSF at pH 7 (liquid-like) showed curves like those reported for the extracted liquid SF from the gland, suggested to be "aquamelts," due to their similarity to typical Maxwellian polymer melts[6,62]. Notably, the cross-over frequency seemed to be shifted towards higher frequencies as pH increased, indicating a shortening in the lifespan of the interaction. Again, at pH lower than 7, the observation of gel phases indicates long-lived and increased stability of the formed structures. In all cases, given that the LVE of the material was not exceeded, samples remained clear, and repeated cycles showed similar behaviour.

Similarly, when measuring viscosity against shear rate for samples at pH 7, the material undergoes shear-thinning until the shear viscosity increases, preceded by a normal force increase, resulting in fibrous aggregates akin to NSF (Fig. S14)[53]. Interestingly, under similar experimental conditions, this transition seems to be hindered at higher pH values (something also seen for native silks and our samples in LiBr[66], in congruence with the dominant liquid character and the decreased stability of the interaction that gives rise to the slowest relaxation mode[67].

The proximity of the critical pH observed here with that reported for NTD (below pH 7)[51], strongly hints at this domain being involved in the observed structural transitions. Furthermore, the transition is also reflected in an earlier onset of aggregation observed at pH lower than 7 when temperature ramps are conducted on CD and DLS, Figs. S15–17. Consequently,

we hypothesise that the absence of NTD would render the material pH-insensitive within the range (pH 8-6) and such a rheological transition would not be seen.

As a means of comparison and in support of our use of NLSF as a model system for NSF, RSF was also studied for its rheological behaviour and response to pH changes. It is well known that silk fibroin behaves very differently after standard regeneration/reconstitution[68,69]. For example, when shear is applied to NSF extracted directly from the posterior part of the middle gland, higher-order assembly into nanofibrils is observed, contrary to RSF at a similar pH of 7[7,70,71]. Similarly, reversible sol-gel transitions have been observed only for NSF when changing pH[58], while our RSF is insensitive to pH in the studied range. It is possible to promote assembly into nanofibrillar structures in RSF, but only when the pH is decreased to near the theoretical isoelectric point (PI) of FibH (ca. 4.7)[72], well below the physiological pH of the gland[68,70,73].

Many authors argue that the molecular weight (MW) reduction in RSF is solely responsible for these observations[74–78]. However, we considered this reduction insufficient to explain the major differences in literature found viscosities[79], or the complete insensitivity to pH (between pH 8 and 6) that we observed. Reported viscosities obtained from NSF are far superior to those reported for RSF even at similar weighted concentrations (between 500–4000 Pa•s for SF and between 0.1–1 Pa•s RSF)[62,80]. Such a large difference, ~4 orders of magnitude, is not substantiated by the MW difference, which on average only falls by approximately half after 30 min of boiling during degumming[78], as depicted in Fig. S18. This reduction in MW would be translated in a proportional reduction of ~0.088 (calculation in Supplementary Note: polymer viscosity), close to one order of magnitude drop in viscosity (3 orders of magnitude away from the difference observed experimentally). Notably, when comparing NLSF and RSF solutions in LiBr (fully denatured) at similar concentrations a drop of this magnitude is observed (Fig. S19).

Thus, we hypothesised that coupled with the reduction in MW, RSF is also losing its NTD, which as noted previously, is highly implicated in the improved rheological properties of NLSF and NSF. A molecular model of the NTD with the first repetitive domain is shown in Fig. 4a. To prove this hypothesis, we conducted proteomics analysis on the peptide fragments (0.1–6 kDa) obtained directly during the dialysis of the liquid solutions of both RSF and NLSF. Interestingly in this procedure whilst substantial material was obtained from RSF, no material was recovered from NLSF. Together, the SDS-PAGE results obtained from NLSF, and this observation would indicate a near-native MW and the likely presence of all its functional domains. On the other hand, LC-MS/MS obtained from the RSF fragments found a total of 113 peptides which matched the provided sequences (FibH, FibL and P25). With 35 matching FibH for 10.17% coverage, 52 peptides FibL for a 96.56% coverage, and 26 matched P25 with total coverage of 89.09% (Fig. S20a–c). For brevity, the following discussion will be focused on FibH, given the predominant role of this protein in the system. However, the high coverage of both FibL and P25 indicates a high degree of hydrolysis in RSf[78]. Of the 35 detected peptides from FibH (Fig. 3b), 22 belong to the first 150 amino acids; 2 of these have a 100% confidence match, with other 3 high confidence matches with 81, 78, and 60% of confidence. The higher number of matched fragments and their higher confidence alludes to preferential cleavages at the NTD (Fig. S20d, e). Although the results do not allow us to unambiguously determine NTD as the sole determinant for the transitions, as P25 and FibL could play a role in assembly, the high degree of NTD homology and presence across all Ditrysia[64,81,82], and the absence of the latter two proteins in Saturniids[19,83,84], reinforces the idea that this domain is critical in driving assembly.

Beyond this observation, the complete sequence coverage plot indicates that most of the matched peptides correspond to the more hydrophilic segments of the chain (Fig. S21a), i.e., hydrolysis during degumming in $Na_2CO_3$ is likely limited by accessibility. Furthermore, besides G, the most abundant residues in FibH, S and T are particularly enriched at the terminal positions suggesting possible intramolecular nucleophilic attack under the alkaline conditions. Other amino acids such as N, K, and E were found at the

terminal position of the peptides. Curiously N, despite its low abundance with only 20 in FibH, was found predominantly at the C-terminal of the fragments, with S and T more abundant at the N-terminal positions (Fig. S21b). Overall, during degumming chain scission occurs mainly at S residues, as suggested by previous studies[76], but other hydrolysis mechanisms are also occurring. The hydrophilic spacers and termini are richer in polar amino acids, often considered better nucleophiles in aqueous conditions. Residues R, H, and K are only found in the terminal domains, and E and D both in the spacers and terminals (Fig. S21).

## Structure and supramolecular ordering

Despite the important rheological differences observed with pH changes for NLSF, we did not observe clear differences in the NMR and CD data in the studied pH range, suggesting that most of the secondary structures of the protein remain unchanged. To better understand the system, we then conducted TEM of NLSF samples at pH 8 and 6, above and below the identified transition. Here we observed that the protein appeared as globules with an apparent size of about 24 ± 8 nm at pH 8, but elongated fibrillar structures were observed at pH 6 (Fig. 4b, c and Fig. S22a, b, for details). Although these fibrillar structures were slightly thicker than the expected solenoid (6 ± 1 nm against ca. 3 nm), acquiring higher resolution images proved difficult given their small size and high sensitivity of the samples to beam damage. The observation of no change in NMR and CD data with pH, but such dramatic morphological change observed in TEM suggests that only small subdomains within FibH might be changing (such as NTD and the flexible linkers). This would allow for extension of the solenoid, which seems to be contained within the globules, suiting the observation in DLS and offering a conciliating explanation between the micelle and liquid crystalline models[85,86]. In contrast, RSF is known to be insensitive in this pH range, only showing featureless globular morphologies[72,87], consistent with the hypothesis that the protein is losing vital structurally functional domains.

Beyond the morphological change, we observed the presence of supramolecular structures resembling bottle-brush structures (Fig. 4f and Fig. S23) similar to those observed previously from *Samia ricini* (*Saturniid*) fibroin[88], and recently in NSf[2]. Based on these observations and the apparent role NTD has on the reversible sol-gel transition, we propose that FibH can form high-order oligomers driven by head (NTD) interactions, making the core of the supramolecular fibre as illustrated in Fig. 4h.

The structure of the NTD solved by X-ray crystallography, a dimer (n2) where each molecule donates two β-hairpins. This structure has a saddle shape, with highly hydrophobic patches on the surfaces, proposed to form tetrameric units (dimer of dimers); herein we refer to this first interface as n2/n2. The remaining β-sheet surfaces are covered by a short α-helixes linked to the rest of the structure by a flexible spacer (Fig. 4d). Given that this helix can flip out of the hydrophobic space[64], we propose that NTD tetrameric units could stack, through what we called tetramer-tetramer interfaces (n4/n4), as proposed in Fig. 4g, leaving the rest of the multidomain protein body protruding laterally. These structures impart great ordering by creating supramolecular brush-like fibres (Fig. 4e, f, h) which might be responsible for the cholesteric textures observed previously[36]. Hence, we conducted docking simulations from NTD crystal units at pH 8, 7 and 6 to verify that these can undergo pH-dependent oligomerisation (Fig. S24a). Our results suggest that NTD likely forms higher-order oligomers, with the tetrameric interactions being strongly favoured at pH below 7 (Fig. S24b–e). At this pH, the protonation of residues Asp44 and Asp89 promote the stability of the interaction; both strictly conserved throughout all FibH in *Ditrysia* and belonging to acidic clusters[82,89]. Moreover, it was noted that all the interfaces (dimer/dimer and tetramer/tetramer), even if slightly different in estimated ΔG values, were in a similar order of magnitude. Current work is undergoing to obtain high resolution experimental data of this transition. Similarly, we noted that solenoid structures would be left in position for lateral interactions to emerge upon oligomerisation. Docking simulations of solenoid units verified that such interactions are possible and equally favourable parallelly or antiparallelly driven by the exposed A

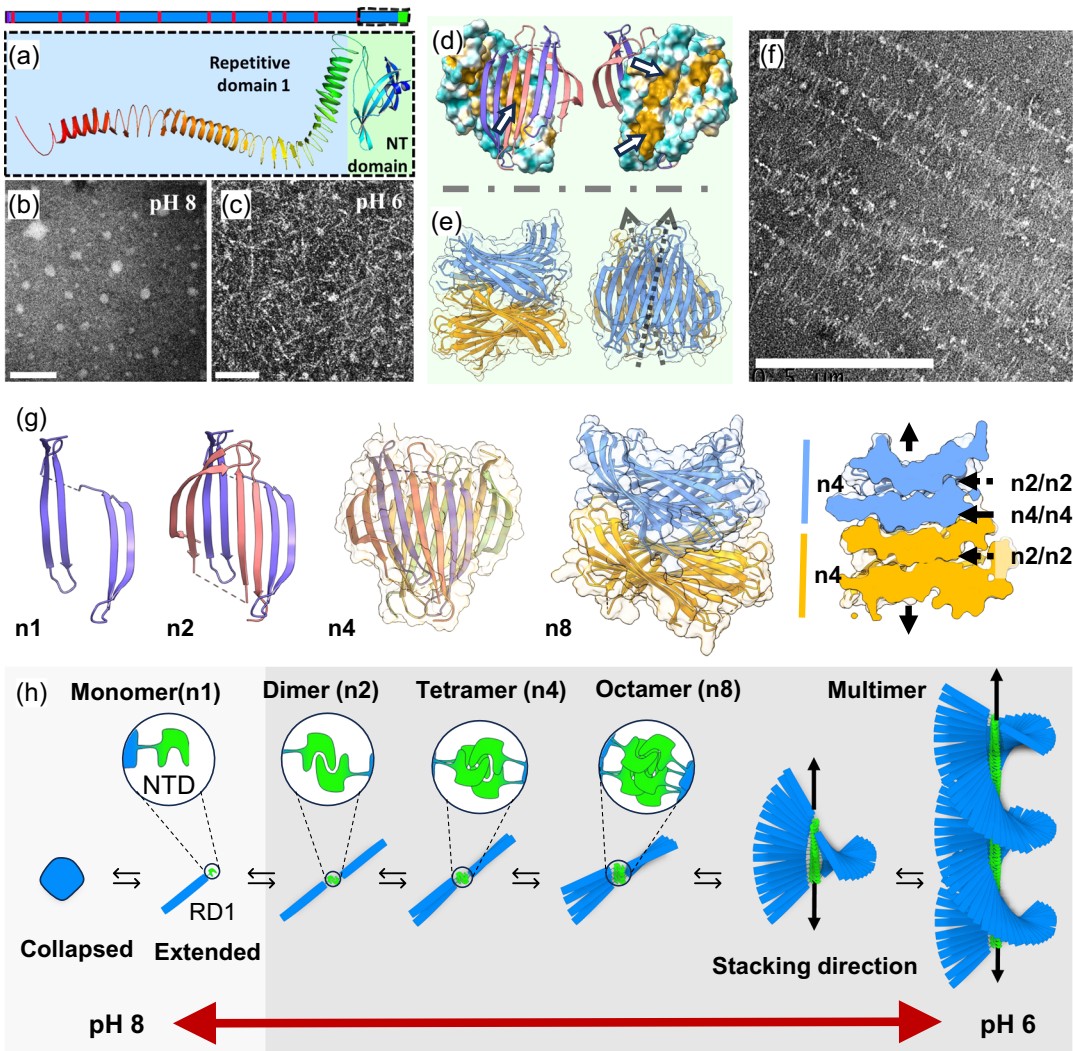

**Fig. 4 | Evidence of a reversible pH-driven assembly in NLSF. a** Cartoon of Fibroin heavy Chain with 12 repetitive blocks (blue), 11 flexible linkers (red), NTD (green) and CTD (purple), under this a molecular cartoon of one of the folded models corresponding to NTD and first repetitive domain in a beta-solenoid type of fold. **b** Negative stained TEM of NLSF at pH 8. **c** Negative stained TEM of NLSF at pH 6. **d** Surface representation of the proposed biological unit of the NTD tetramer[64] coloured by hydrophobicity (cyan hydrophytic and wheat hydrophobic) with top dimer represented as a cartoon (left) and as seen from the back (right) with white arrows pointing at hydrophobic patches involved in dimer-dimer (left) and tetramer-tetramer (right). **e** Proposed NTD tetramer stacking shows 2 tetrameric units stacked from the side (left) and the top (right) with arrows indication the twist of the stacking. **f** Negative stained TEM of NLSF at pH 6 showing supramolecular brush-like structures formed via NTD stacking. **g** Molecular model of the transition, left to right, from monomer (n1), dimer (n2), tetramer (n4) and octamer (n8) coloured by chain (n1-4), and by tetrameric unit (n8) and a cross-section of n8 indicating the dimer-dimer (n2/n2) and tetramer-tetramer (n4/n4) interfaces by discontinuous arrows and the stacking direction by continuous arrows on the far right; n4 and n8 models also show a ghostly representation of their surface. **h** Proposed pH driven self-assembly of the protein, from globular unimer, extended conformation, dimer (each molecule donates two β-hairpins that interlock), tetramers (as shown in D, formed by dimer units stacking with a central symmetry plane), octamer (as shown in E, formed by the stacking of tetrameric units), and multimeric bottlebrush-like fibril showing proposed cholesteric order (left to right). Scale bars 100 nm for b and c, and 500 nm for f.

residues on the surface (Fig. S25), providing a foci for network formation (Fig. S26).

These structures likely fulfil an early role in the assembly pathway, given the conservation of the NTD domain across all species. However, no evidence of these bottle-brush structures can be found within the fibre (Figs. S27 and S28) suggesting that these may exist transiently. NTD-driven oligomerisation facilitates controlled phase separation, thus aiding in the dehydration of the dope and fostering lateral interaction of the solenoid units. Upon acidification, these structures align with the flow generating the cholesteric textures. Here, extensive lateral interaction forms a network. Thus, in these first steps, the system is characterised by two main interactions, NTD-NTD and solenoid-solenoid (the absence of CTD and FibL in *Saturniids* indicates that these are not essential for fibre formation, as has been suggested recently)[90]. As the silk duct diameter is reduced, a critical

stress is reached, and NTD interactions are disrupted, leaving a network dominated entirely by lateral solenoid interactions (Fig. S29). The order imparted by the NTD oligomers is lost, and so the network can undergo a reorganisation, with the solenoid axis aligning with the flow direction and giving rise to the recently observed fractal network, both in NSF and high-quality RSf[85]. Figure 5 summarises the proposed fibre formation model, with illustrations and experimental microscopy data showing snapshots of the observed structures and correlation with the optical textures.

The described transition explains the transformation of cholesteric-isotropic-nematic observed before[34]. Moreover, the last step leaves the solenoid axis, and thus the hydrogen bonding network, parallel with the direction of elongation and ideally placed for unfolding and stretching of the backbone. Similar denaturation of α-helix fold of fibroin in 1,1,1,3,3,3-hexafluoro-2-propanol (HFIP) under stretching has been observed recently,

**Fig. 5 | Summary of the proposed assembly pathway for fibroin into the silk fibre. a** Cartoon with the schematic representation of the proposed self-assembly pathway for fibroin. In the first instance, fibroin at pH greater than 7 exists as a globular unit (**b**). At pH lower than 7 extend and form supramolecular brush-like fibrillar structures that would align with the direction of the flow and originate a cholesteric texture (**c**). At critical stress, the supramolecular structures break (**d**), and the formed solenoid network aligns under flow to generate the nematic order (**e**, **f**). Scale bars 100 nm in **b**, **d**, **f** and **g**, 500 nm in e. Illustration with optical textures was reprinted (adapted) with permission from Asakura et al. Some Observations on the Structure and Function of the Spinning apparatus in the Silkworm Bombyx mori, 2007 Biomacromolecules, ACS. Copyright 2007 American Chemical Society[38].

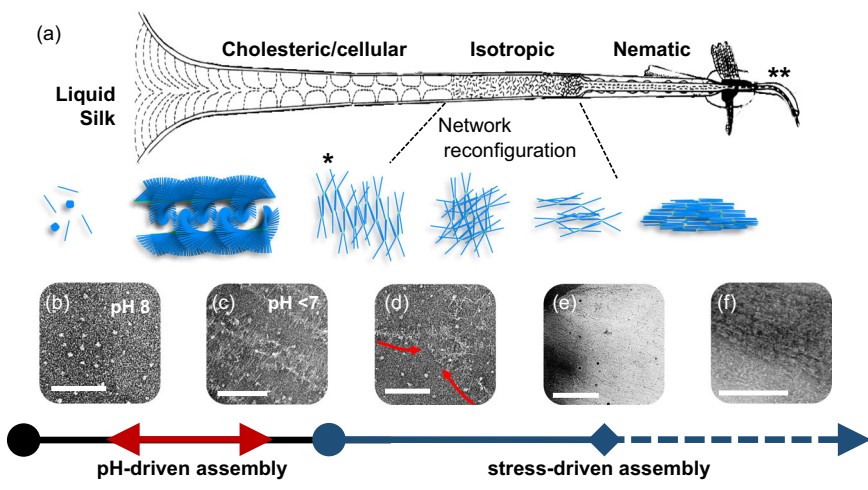

with a α-to-β transformation prompted by stretching[91]. These hypotheses are supported by observing two maxima in yield stress in stress-controlled experiments on gel-like material at pH 7 (Fig. S30). Similar stress maxima were recently observed when pulling fibres from other native-like fibroin solutions[92,93]. The first yield occurs at relatively low strains and appears narrower than the second. Such a double yield behaviour suggests the disruption of two different interactions, which we believe correspond to the initial disruption of NTD interactions, with narrower energy distribution, and secondly, a stress-induced denaturation (hydrogen bond breakage) with subsequent backbone stretching, similar to the critical amount of work observed for fibrillation[77]. The described process leads to the extension of the backbone and β-sheet formation with "nano-fishnet" architecture proposed before[92,94,95]. Based on this hypothesis, we would predict that if the stress is kept low enough during fibre formation, one would potentially be able to obtain Silk-I based fibres (or a combination of polymorphs), in agreement with force reeling work the observation that the rate of work, and not the amount of work in itself are essential to promote the transition to solidification[60,96–98].

## Conclusions

This study employed a comprehensive array of techniques to enhance our understanding of the Silk-I structural polymorph of silk fibroin. Through our investigations, we made meaningful advances in understanding the in-solution structure of FibH and uncovered a pH-induced reversible sol-gel transition driven by NTD interactions, which we subsequently correlated with morphological and structural alterations of the protein at the molecular and supramolecular level.

We propose that FibH from *B. mori* is a multidomain protein with twelve low complexity domains with the ability to fold into β-solenoids. We accredit the Silk-I diffraction pattern to these domains, which are linked by flexible spacers and flanked by two terminal domains, of which NTD is the main driver of supramolecular assembly. As the pH drops below 7, NTD oligomerises forming large bottle-brush supramolecular fibres. The solenoid units, anchored within the construct are enabled to form stable lateral interactions driven by hydrophobic Ala-rich surfaces. Stress in the first instance disrupts the initial bottle-brush structures, leaving behind a fractal network of solenoids interacting laterally. As stress increases, the fold is denatured and the backbone of the polypeptide is stretched in the direction of the flow, driving the assembly into a nano-fishnet molecular architecture with β-sheet crystallites as nodes[94,99]. On the other hand, our study on standard RSF showed that these have largely lost NTD, FibL and P25. Thus, it contains mainly a polydisperse mixture of the low complexity domains of the protein. In the absence of NTD, these fragments are devoid of the pH switch, and therefore lack the degree of preorganisation, albeit still retaining most of the Silk-I features. Our mass spectrometry analysis of proteolysis can therefore provide a powerful method to assess feedstock quality.

Although the β-solenoid structure might be common among silk proteins, we believe it to be incidental, as other fibrillar structures are compatible with the proposed assembly mechanism. NTD being the only essential feature in pH-driven assembly. In silk-spinning *Samia ricini*, the repetitive domains are rich in polyA motifs folding in α-helixes[100,101], and also form the bottle-brush structures[88]. Therefore, our model provides a general description for silk-fibre formation independent of specific sequences. Beyond insect silk, our work indicates that other low complexity proteins, especially those rich in GX motifs, such as the class III of G-rich proteins and other non-characterised fibroin heavy chain-like proteins widely found across all kingdoms of life. We also note the uncanny resemblance of FibH with known toxic GA repeat proteins associated with disease (e.g., amyotrophic lateral sclerosis, ALS), which in the monomeric species might have a similar fold, and their aggregation pathway might be similar as the proposed here; it has already been proven that $(AG)_{15}$ peptides show similar conformation to FibH in the solid state[52,54,102,103].

In a wider context, our work opens the opportunity for further interrogating the structure of simple repeat proteins, associated both with health and in disease development. Many of these are believed to be disordered, but superstructures of polyglycine-II or polyproline-II helixes might underpin their function. Within the silk community, our work also paves the way for the generation of materials based on native-like silk fibroins. Overall, through our findings, we provide a unifying model that accounts for the observations on silk-fibre formation made through decades of research on the fibroin assembly process, from a liquid state to the remarkable structural material that constitutes silk fibre and might serve as inspiration for the design of bioinspired materials.

## Materials and methods
### Native-like silk fibroin solution (NL-SF)

Fibroin solutions of 8.5 M LiBr and 20 wt% of protein were kindly provided by Orthox LTD in 60 mL syringes and stored at 4 °C until used. Briefly, silk fibres were initially treated as per patent number US 8,128,984B2 using mild degumming conditions to minimize fibroin hydrolysis, after washing with MilliQ water, fibres were gently dried and dissolved in 8.5 M LiBr at 60 °C over 4 h. A stock solution at 2 wt% was made for studies requiring diluted solutions by diluting the provided solution with 8.3 M LiBr. The 20 wt% solutions look clear, free of precipitates, with a distinctive pale-yellow tint and low-shear viscosity of 60 ± 3 Pa.s. The stock solutions were then dialysed against MilliQ water using 12–14 kDa molecular weight cut-off (MWCO) dialysis membranes, with regular medium- to fresh MilliQ water changes. LiBr removal was followed by conductivity measurements of the dialysate and contrasted against a conductivity calibration curve, often requiring 3–4 days to remove 99.9% of LiBr salt. After dialysis of the 20 wt% solution, a clear gel is obtained. In contrast, the 2 wt% yields a clear liquid solution with viscosity like pure water. Concentrations were

determined gravimetrically by cutting out small fragments or aliquoting about 500 μL of the solution and lyophilising them overnight in a Labconco Freezone 2.5 L Freeze-drier. The high concentration solution produces a gel with concentrations in the range of 60–80 mg/mL, whereas the 2 wt% solution produces a solution in the range of 5–7 mg/mL.

## Standard Regenerated silk fibroin solution (RSF)

A standard regeneration protocol was followed[37]. Briefly, about 10 g of raw silk, kindly provided by Orthox LTD, were boiled for 30 min in 4 L of aqueous 0.02 M $Na_2CO_3$, the supernatant was then discarded, and the fibres were rinsed with copious MilliiQ water at room temperature. The wet degummed fibres were dried in a convection oven at 30 °C overnight. The dry degummed fibres were weighted to verify degumming efficiency. Finally, degummed fibres were dissolved by adding specified volumes of 9.3 M LiBr to obtain a 20 wt% silk fibroin solution and left in an oven at 60 °C for 4 h, or until total dissolution was observed (whichever occurred first). The solution was then centrifuged at 5000 units of relative centrifugal force (RCF) for 5 min at room temperature to remove small undissolved particles. After these steps, the solution looks clear, free of precipitates with similar pale-yellow tint and with low-shear viscosity of 2 ± 0.5 Pa.s. The solutions were then dialysed against MilliQ water following an identical procedure as before. After dialysis, the 20 wt% solution produces a clear solution with a viscosity not different to pure water. Concentration was determined gravimetrically by aliquoting 500 μL of solution and lyophilising it overnight. The 20 wt% LiBr solution produces an aqueous solution with 60–80 mg/mL. Given the liquid nature of this sample, dilutions were prepared directly from the concentrated stock.

## Sodium Dodecyl sulphate Polyacrylamide Gel Electrophoresis (SDS-PAGE)

For SDS-PAGE measurements, Tris-acetate 8–12% gels were used from Sigma. Samples of 0.1 mg/mL of NL-SF and RSF were used by mixing with stock solutions of running buffer and Coomassie blue. A high molecular weight ladder, HiMark Pre-stained Protein ladder from Invitrogen was used. Gels were run as recommended, using 200 V and 1 Amp for 40 min. Gel images were then analysed using FIJI by measuring grey-scale values and plotting coaligned with the standard ladder.

## Rheology

The rheological characterisation was done in a Malvern Kinexus Pro rheometer with a Peltier lower plate for temperature control and with either a 20 mm parallel plate (PP20) upper geometry for gel samples, or with a truncated conical geometry with 4 degrees angle and 40 mm in diameter (CP4/40) for liquid samples. Samples were equilibrated for 24 h at room temperature in x50-100 volume of 100 mM Tris buffer at indicated pH values, titrated using stock solutions of 2 M HCl or 5 M NaOH. Although the buffering efficacy of Tris at pH 6 is very low, this buffer is preferred as other common buffers such as phosphates reduces the storage time considerably, promoting early conformational changes to the NL-SF concentrated solutions/gels.

Briefly, 500 mL of buffer solution were prepared at the desired pH. About 5–10 mL of aqueous SF solution were placed in them, contained by similar dialysis tubing membrane from previous steps. Samples were then left under slow stirring (ca. 150 rpm) at room temperature for 24 h to equilibrate. Following this, samples were taken out from the buffer solution and either carefully poured into clean 50 mL conical end centrifuge falcon tubes for liquid samples or taken to standard plastic Petri dishes (100 mm diameter) for gel samples. These last samples were taken out of the dialysis tube by carefully cutting open the membrane with a clean new scalpel. Samples were used immediately afterwards.

The following procedures were adapted from different works in the literature that work with native silk fibroin solutions, with minor modifications. In every case, samples were treated with the utmost care to prevent early conformational changes prompted by poor handling. Gel phase samples were carefully collected by excising small fragments from the bulk

using a wet scalpel and disposing of any sample that would show evidence of aggregation (opaque fibrillar aggregates). Enough gel sample was used to fill the entire geometry, carefully positioned in the centre of the lower plate, with the upper plate lowered slowly with a maximum allowed normal force of 1 N to a 1 mm gap. Water saturated tissue was left around the geometry, making sure it was not in contact with it, and the entire geometry was enclosed by a plastic cover to prevent dehydration during experiments. For liquid samples, a 5 mL pipette with tips with cut ends were used to prevent applying high shears when handling, with a standard quick loading setting from the software and similar procedures to prevent dehydration.

For the determination of the Linear Viscoelastic Limit (LVL), both elastic (G') and viscous (G") moduli were measured in amplitude sweeps done in oscillatory mode at a constant frequency of 1 Hz, in the strain range between 0.1 and 100%, with 10 points per decade. Measurements at each point were taken until convergence was detected by the Kinexus software. Liquid samples underwent a pre-homogenisation step, as proposed elsewhere[60,100], by conducting fully rotational low-shear rate (1 s$^{-1}$) deformation for 100 s. At least 3 repeats were done for every sample.

After determination of LVL, G' and G" were measured under frequency sweeps on fresh samples at constant strain of 0.01 (1%, within LVL of all samples) between 10 and 0.1 Hz. Similarly, 10 points per decade were measured, with measurements being taken until data convergence was observed by the software. Liquid samples underwent homogenisation, as mentioned before, by applying fully rotational deformation at 1 s$^{-1}$ for 100 s before conducting the experiments. At least five repeats were done for every sample.

Only liquid samples were used for viscometry. Viscosity against shear rate measurements was obtained by a shear rate table in the range between 0.1 and 100 s$^{-1}$, with 10 points per decade. Measurements were taken until convergence was obtained at each shear rate point. Shear viscosity, as well as normal force values, were recorded.

## Fibre X-ray diffraction

These experiments were done using Rigaku rotating anode (Cu Kα) with Saturn CCD detector with exposure times of 30–60 s and specimen to detector distance of 50 or 100 mm. Reflections were measured using CLEARER software.

For diffraction obtained from degummed silk fibres, about 1 cm long fibre bundles were mounted on glass capillary tubes using epoxy-based glue and then mounted directly onto a goniometer head for alignment. Alignment was ensured by monitoring a microscope camera focused on the beam path, and alignment was adjusted for the nominal 0 and 90° orientations. Spectra were acquired at 50 and 100 mm of distance at 0 and 90° orientations, for 30 and 60 s, respectively.

The silk-I film was fabricated by slow drying about 100 μL of NLSF solution at 6 mg/mL at room temperature, drop-casted onto a clean petri dish surface. Once dried, the film was carefully removed using forceps, glued to a capillary glass tube, and observed as the fibres before. Here, the film was orientated so the beam would strike normal to the surface at 0°, parallel to the film surface at 90° and later a range of acquisitions were done by varying orientation by 5° between 0° and 90°.

## Liquid chromatography-mass spectrometry (LC-MS/MS) and proteomics analysis

Samples were analysed following modified Shotgun proteomics protocols. For the analysis of the hydrolysis patterns, concentrated solutions in LiBr of both samples were dialysed against MilliQ water while being contained within a dialysis tubing membrane with 6 kDa MWCO, which in turn was contained by a 100 Da dialysis tubing membrane. Care was taken not to cross-contaminate the different spaces, and the space between membranes was filled with 15 mL of MilliQ water. After 99.99% of LiBr was removed by dialysis, determined by the conductivity of an 8 h dialysate from fresh, the samples collected from the volume between the 6 kDa and 100 Da membranes were transferred to sterile 50 mL conical centrifuge falcon tubes, from which samples were directly summited to be analysed. Obtained

sample from RSF contained about 2.0 ± 0.5 mg/mL of peptides, but no material was recovered from NL-SF (<0.1 mg/mL).

Samples were injected in a Dionex RS3000 High pressure Liquid Chromatographer (HPLC), coupled with a Orbitrap Elite in Electrospray ionisation mode (ESI) to analyse the fragmentation of the submitted peptides by ESI-MS/MS.

The collected data was then analysed using PeptideShaker and contrasted against the sequence of FibH, FibL, and P25 from the UniProt database (accession P05790, P21828, and P04148, respectively).

### Nuclear magnetic resonance (NMR)

For NMR studies, only samples at pH 8 and 6 of NL-SF were studied. Briefly, stock 2 wt% solution was dialysed against MilliQ water until >99.9% of LiBr was removed to obtain a solution of about 1 mg/mL of protein. These solutions were then passed through a NAP45 non-interacting column, following manufacturer instructions, pre-equilibrated with 10 mM Tris in deuterium oxide, titrated beforehand to the desired pH with aqueous 2 M HCl or 5 M NaOH. The collected buffered samples were then analysed using a Bruker Avance III HD 700 with a 1.7 mm cryo-enhanced probe. $^1$H-NMR were collected, as well as proton-proton Total Correlation Spectroscopy ($^1$H-$^1$H TOCSY) for residue assignment, Nuclear Overhauser Effect Spectroscopy (NOESY) for distance determination and natural abundancy carbon-proton Heteronuclear Single Quantum Coherence ($^1$H-$^{13}$C HSQC) for secondary structure determination of A and G residues.

### Fold simulations and model generation

Fold simulations were carried over at two different servers to verify results congruency. However, in general, the AlphaFold2 prediction methodology was used. The services used were those published on Google Colab books by DeepMind and ColabFold teams[41,42]. For the simulations, Colab Pro, was used to ensure higher assigned memories, better processors and longer runtimes.

**Fibroin heavy chain fold simulation.** Given the detected modularity of the multidomain protein and the size limitation on the simulations through ColabFold (1000–1400 residues), only the NTD and the first repetitive domain were used initially for a total of 650 amino acids. The multisequence alignment (MSA) was run using Jackhammer and mmSeqs2, with a minimum pair coverage with a query of 50% and minimum sequence identity of 20%. Jackhammer offered the best coverage overall, with 7456 Sequences found in Uniref90, 9062 Sequences found in SmallBFD, and 482 Sequences found in Mgnify, or 17000 sequences in total; most of these covered the repetitive domain. Next, Alphafold was run, and five models were generated after running the trunk of the AlphaFold neural network 8 times with different random choices across the MSA. Finally, the generated five models were relaxed using Ammber-relax forcefield, and the models were downloaded, visualized and post-processed using Pymol. On the other hand, mmSeqs2 found about 45 sequences, primarily for NTD, and the repetitive domain was modelled as a random coil.

Later, the same sequence was also run on the DeepMind colab book, and very similar models were obtained. Following these, the rest of the domains were modelled in fragments below 1000 residues that contained entire domains (either repetitive domains alone, in combination with one linker, or as two repetitive domains with a single linker). Models obtained through this method were used to generate the full FibH structure.

### Docking simulations

**NTD docking.** In the first instance, the solved N-terminal tetramer model (PDB: 3UA0) was relaxed using Rosetta. Then, using the relaxed model, an octameric (tetramer-tetramer) stacked unit was generated with manually adjustments, and the model was further relaxed as an octamer.

Docking simulations under pH control were done by first determining the protonation state of charged residues on the relaxed octamer model.

Next, protonation states were predicted by PDB2QR server (https://server.poissonboltzmann.org/pdb2pqr) at pH 6, 7 and 8. These generated models were used as references for root-mean-square deviation (RMSD) analysis during docking experiments. Finally, local docking was run in Rosetta on each of the three models, generating 2000 independent simulations for each. Estimated free energy of interaction measured in Rosetta energy units (REU), ΔΔG, was then plotted against RMSD (a measure of the deviation from reference models).

**Solenoid lateral dockings.** Models obtained from the first repetitive domain was used here. It was noted that the first repetitive domain could be divided into 3 sub domains linked by less regular solenoidal structures that are thought to allow for flexibility within the repetitive domain. Furthermore, the first and second subdomains were unique to the first repetitive domain, whereas the third subdomain was more represented across all repetitive domains. Hints at this were already observed after self-similarity analysis. Consequently, the model was separated into three sliced models containing the subdomains, each relaxed in Rosetta previous to global docking simulations, where only models with ΔΔG lower than -30 REU were used for further analysis.

### Far UV circular dichroism (CD)

CD measurements were performed using low concentrations of protein (ca. 0.05 mg/mL) in 10 mM sodium phosphate buffer at the desired pH (6–9) at a temperature of 25 °C. Buffer pH value was adjusted with aqueous dilutions of phosphoric acid or NaOH. Samples from NL-SF were obtained by dialysing stock 2 wt% solution in 8.3 M LiBr and equilibrated in x100 volume of buffer over 24 h. After equilibration, samples were centrifuged at 6000 RCF for 10 min at room temperature to remove any precipitate and later diluted with fresh buffer until obtaining High-tension (HT) values under 700 for the whole studied range. Spectra were acquired from 240 to 190 nm in a 1-mm path length quartz cuvette using a JASCO J-820 spectropolarimeter. The wavelength step for the CD measurements was 1 nm, and the scan rate was 100 nm/min. The shown spectra were background corrected and averaged over five scans. For temperature analysis, similar procedures were followed, the sample was heated in 5 °C steps from 25 to 95 °C and equilibrated for 30 s at each temperature.

### Dynamic light scattering (DLS)

DLS measurements were performed using 1 mg/mL protein solutions. After dialysing the 2 wt% in 8.3 M LiBr solution, samples are collected with approximate concentrations of 1 mg/mL, equilibrated in buffer at desired pH (10 mM sodium Phosphate) over 24 h and used directly after centrifugation at 6000 RCF for 10 min. Disposable PS large volume cuvettes were used in a Malvern Instruments ZetaSizer nano ZS where data was acquired using software that included macros for size analysis and temperature trends. Software provided materials parameter values for protein sample, and water for media was used. For temperature trends, samples were equilibrated for 60 s at each temperature, 3 scans were averaged, and the temperature ranges from 25 to 75 °C with 1 °C increment.

### Transmission electron microscopy (TEM)

Jeol JEM 1200 with a tungsten filament and acceleration voltage of 120 kV and a Jeol JEM 2100 with a LaB6 electron gun at 200 kV. Both were used for brightfield and negatively stained images.

Tecnai T20 with a LaB6 filament and acceleration voltage of 200 kV for negatively stained samples, and a Tecnai F20 with a field emission gun (FEG) with an acceleration voltage of 200 kV for cryo-TEM. For the latter, a Vitrobot Leica EM GP with humidity set to 100%, blotting time 1 s, was used after adding 5 µL of the sample before plunging into liquid ethane, and grids transferred to a cryo-holder (Gatan, Inc.).

The higher resolution cryo-TEM were taken using a Talos Arctica equipped with a 200 kV X-FEG, Ceta 16 M CCD detector, Gatan K2 DED, and Gatan GIF Quantum LS energy filter. Samples were vitrified in this case using a FEI Vitrobot.

## Statistics and reproducibility

We did not observe batch-to-batch variability when using NLSF, and thus, we consider all our replicates as technical replicates. All data was collected in, at least, triplicates (or otherwise stated in the legend) and results are reported as the average with their corresponding standard deviation when applicable.

## Data availability

All data are available in the main text or the supplementary information. The source data behind the graphs in the paper can be found in Supplementary Data 1. Additionally, raw data files for NMR, MS, fibre XRD and AlphaFold (.cif) are available at the University of Bristol data repository, data.bris, at https://doi.org/10.5523/bris.2rn8nb1bualvs2sphf60f5wfy7.

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

## Acknowledgements
The authors would like to express their gratitude to the following individuals and facilities for their contributions and assistance: J.C. Eloi from the Chemistry Imaging Facility, J. Mantell from the Wolfson Bioimaging Facility, U. Borucu from the GW4 Cryo-EM Facility for their help and insightful contribution in TEM; M. Crump, and specially C. Williams for their assistance and discussions on protein NMR, and access to the 700 MHz spectrometer; C. Arthur for contributions and assistance with LC-MS/MS; R. Cruz-Samperio for help with SDS-PAGE and discussions; P. Laity for contributions and thoughtful discussions related to this manuscript. The authors would also like to express their gratitude for and acknowledge the funder contributions: EPSRC National Productivity Investment Fund grant EP/R51245/XF (R.O.M.T.), EPSRC Doctoral Prize Fellowship at the University of Bristol grant EP/W524414/1 (R.O.M.T.), Wellcome Trust grants 086906/Z/08/Z, and 100917/Z/13/Z (N.S. and R.W.), The EIC Accelerator grant 947454 (N.S. and R.W.), the NIHR i4i Invention for Innovation award II-LB-0417-20005 (N.S. and R.W.), EPSRC early career fellowship grant EP/S017542/1 (F.P.), EPSRC grants EP/K035746/1 and EP/M028216/1 (TEM) and EP/X015416/1 (C.H.), Wellcome Trust grants 202904/Z/16/Z and 206181/Z/17/Z (TEM), BBSRC grant BB/R000484/1 (TEM), BrisSynBio, a BBSRC/EPSRC Synthetic Biology Research Centre, grant BB/L01386X/1 (NMR), BBSRC Alert 20, grant BB/V019163/1(NMR) and China Scholarship Council (Y.L.). The views expressed in this work are those of the author(s) and do not necessarily reflect those of the NIHR, the Department of Health and Social Care or any of their funding bodies.

## Author contributions
Conceptualization: R.O.M.T., S.A.D., L.S., C.H., N.S., and R.W.; Methodology: R.O.M.T., S.A.D., C.H., and L.S.; Investigation: R.O.M.T., L.S., and Y.L.; Visualization: R.O.M.T. and Y.L.; Funding acquisition: S.A.D., N.S., R.W., and R.O.M.T.; Project administration: R.O.M.T. and S.A.D.; Supervision: S.A.D., N.S., and R.W.; Resources: N.S., R.W., Y.L., F.P., and R.O.M.T. Writing – original draft: R.O.M.T. and S.A.D. Writing – review & editing: R.O.M.T., Y.L., F.P., N.S., R.W., L.S., C.H., and S.A.D.

## Competing interests
The authors declare no competing interests.
