## [Peer Review File · Communications Biology]

Reviewers' comments:

Reviewer #1 (Remarks to the Author):

The manuscript “Silk road revealed: mechanism of silk fibre formation in *Bombyx mori*” by Moreno-Tortolero et al. may be acceptable for publication in *Communications Biology* after minor revision.

The authors present a comprehensive study on the rheology of fibroin solutions, as well as on the microstructures obtained from these solutions. The number of characterization techniques is extensive and the results are convincing, so that some aspects of the spinning process of *Bombyx mori* silk seem to be clarified from this work. In this regard, I have just a couple of suggestions to enhance the readability of the manuscript.

The title seems slightly misleading in its present form. I would suggest a more informative title, such as “Molecular organization of fibroin heavy chain and the mechanism of silk fibre formation in *Bombyx mori*” or similar.

Page 5, Line 172. In addition to the references mentioned after the sentence “It has largely been recognised that in native spinning both shear stress and pH changes are responsible for the transformation of the soluble fibroin into the solid fibre”, and due to the central role of this hypothesis on the spinning of *B. mori* silk, the following reference is relevant: Ceniz et al. *Soft Matter* 11 (2015) 8981. This work is a clear evidence that it is possible to obtain a silk fiber by simply immersing the silk gland in a mild acidic solution and stretching it. Consequently, no additional requirement is necessary for the fluid to solid transition during spinning.

Page 8, Figure 4. This Figure is critical to all the work and, consequently, it should be absolutely clear to the reader. In this sense, Figure 4 should reflect the model proposed by the authors with respect to the supramolecular organization of the proteins. In its present form, it is not sufficiently clear the transition from the single molecule to the tetramer and especially, between the tetramer and the multimer. In particular, the geometry of individual proteins should be clearly shown in the multimer, including the situation of selected N-terminal domains and some solenoids.

Additionally, I suggest the following minor amendments:

- Page 2, Line 60. Silk-1 -> Silk-I

- Page 7, Line 227. Carton -> Cartoon

Reviewer #2 (Remarks to the Author):

The review of this paper is as follows, and the reviewer completely disagree with the Silk I and Silk II structural models proposed in the paper for the following reasons.

In this paper, using the Alphafold2 simulation of the repetitive domain of FibH, the authors found that these repetitive domains are predicted to adopt a β -solenoid conformation. They said that the predictions showed low confidence, and relatively high predicted Local Distance Difference Test (pLDDT) in average of about 50.

Namely, this is an initial prediction with low confidence, and further experimental verification is clearly needed. This result is an important step in understanding the structure and function of proteins because the proposal of the structural model should be done very carefully using many experimental data about Silk I.

However, their experimental data to determine Silk I structure in molecular level used by them is quite few, only simple X-ray diffraction data and solution NMR chemical shift data. Therefore, more reported experimental data should be used to check the model.

In the process to propose type II b-turn model for Silk I, Asakura et al. used many experimental data and therefore, the b-solenoid model should be checked by comparing with these experimental data.

1. Asakura et al. used many REDOR experimental data. Many ^{13}C ^{15}N distance informations were obtained experimentally and calculated for type II b-turn model. The agreement between the observed and calculated distances was excellent as summarized in Table 1 (T.Asakura et al. *Macromolecules*, 2005, 38, 7397-7403).

Table 1 and Fig.

T. Gullion et al. (Gullion and Asakura et al. *JACS*, 2003, 125, 7510-7511) also tried $^{13}\text{C}^2\text{HREDOR}$ experiment and supported type II b-turn structure of Silk I.

The calculated ^{13}C .. ^{15}N distances for b-solenoid model using the torsion angles reported in Fig.S10 should check the agreement with the observed many distance data.

2. The type II b-turn model can explain the observed (Iobs) and calculated (Icald) structure amplitudes using X-ray diffraction data of Silk I film very well (T.Asakura et al. *Macromolecules*, 2005, 38, 7397-7403).

Table

The authors should try these checks using their X-ray diffraction data of Silk I.

3. The another experimental data are ^1H DQMAS NMR data of Silk I although they did not cite in their paper. (T. Asakura et al., Determination of Accurate ^1H Positions of (Ala-Gly) $_n$ as a Sequential Peptide Model of Bombyx Mori Silk Fibroin before Spinning (Silk I). *Macromolecules* 2013, 46, 8046–8050).

In the paper, Asakura et al. described as follows.

The accurate ^1H positions of alanine-glycine alternating copolypeptide, (AG) $_{15}$ with Silk I structure were determined. For the purpose, the geometry optimization was performed starting with the atomic coordinates of the hetero atoms reported previously (T. Asakura et al. *Macromolecules* 2005, 38, 7397–7403, *J. Mol. Biol.* 2001, 306, 291-305) and applied only for protons under periodic boundary conditions. The agreement between the calculated and observed chemical shifts of all ^1H , ^{13}C and ^{15}N nuclei was excellent, indicating strongly that the determination of all the atomic-coordinate including ^1H nuclei was performed with high accuracy. Here the ^1H chemical shift was obtained by using both 1 mm microcoil MAS NMR probe-head for mass-limited solid-state samples developed by Asakura et al. and ultrahigh field NMR at 920 MHz. The DQ correlations in the ^1H DQMAS NMR spectra were also used to confirm the intra and intermolecular structures obtained here.

The authors should check the b-solenoid model using these experimental data.

Two Figs.

4. The solution NMR results of Silk I (Suzuki and Asakura et al. *Biomacromolecules*, 2014, 15, 104-112) also proposed type II b-turn structure. This was obtained from the observed chemical shifts ($^{13}\text{C}_\alpha$, $^{13}\text{C}_\beta$, $^{13}\text{C}_\text{O}$, $^1\text{H}_\alpha$, $^1\text{H}_\text{N}$ and ^{15}N) of the backbone atoms using TALOS-N. The torsion angle of Gly residues (77° , 10°) was obtained from solution NMR chemical shifts as well as the 2D diffusion NMR and REDOR experiments of solid-state NMR with very careful manner.

About Silk II model, T. Asakura et al. reported recently from the following references that Silk II is a lamellar structure.

(1. Asakura, T.; Aoki, A.; Komatsu, K.; Ito, C.; Suzuki, I.; Naito, A.; Kaji, H. Lamellar Structure in Alanine–Glycine Copolypeptides Studied by Solid-State NMR Spectroscopy: A Model for the Crystalline Domain of Bombyx Mori Silk Fibroin in Silk II Form. *Biomacromolecules* 2020, 21, 3102–3111.

2. Asakura, T.; Ogawa, T.; Naito, A.; Williamson, M. P. Chain-Folded Lamellar Structure and Dynamics of the Crystalline Fraction of Bombyx Mori Silk Fibroin and of (Ala-Gly-Ser-Gly-Ala-Gly) $_n$ Model Peptides. *Int. J. Biol. Macromol.* 2020, 164, 3974–3983.

3. Asakura, T.; Williamson, M. P. A review on the structure of Bombyx mori silk fibroin fiber studied using solid-state NMR: An antipolar lamella with an 8-residue repeat, *Int. J.*

Biol. Macromol. 2023, 245, 125537.

4. Suzuki, Y.; Morie, S.; Okamura, H.; Asakura, T.; Naito, A. Real time monitoring of the structural transition of Bombyx mori liquid silk under pressure by solid-state NMR, J. Am. Chem. Soc. 2023, 145, 22925-22933.)

In addition, the 1H–1H distance information obtained from DQMAS 1H NMR clearly demonstrates that Silk II is antipolar, not polar as follows. This is different from the Silk II model (polar stem) proposed by authors.

In order to distinguish between these models, antipolar or polar, Asakura et al. obtained 1H–1H distance information from double-quantum magic angle spinning (DQMAS) 1H NMR spectra. In these spectra, only 1H–1H distances within about 4 Å give rise to observable cross-peaks. A set of nine 1H–1H correlation signals from the 1H DQMAS NMR spectrum of (AG)₁₅ is marked in Fig. 6 (Ia). These signals were compared to the distances calculated from the polar Marsh and antipolar Takahashi models. Fig. 6(II) shows that the Gly H α 2 protons in adjacent β -strands are very close to one another (about 2 Å) in the polar Marsh model and therefore a diagonal peak for Gly H α 2 should be detected. However, in the anti-polar model this distance is more than 4 Å, and therefore no peak is expected. This feature is difficult to distinguish from Fig. 6(Ia) because the Gly H α 2 and Ala H α peaks have overlapping chemical shifts. Asakura et al. therefore synthesized deuterium-labeled ([2-d]AG)₁₅ and acquired another DQMAS 1H NMR spectrum (Fig. 6(Ib)). There is clearly no Gly H α 2 peak on the diagonal, now that the Ala H α signal at around 5.0 ppm has been removed. This observation provides very strong evidence that the polar model cannot be correct. Other experimental 1H–1H distance data were also only compatible with the antipolar model. For example, the observed cross-peak (v) is between Gly H α 2 and Ala H α which is longer than 4 Å in the polar model, but about 2.4 Å in the antipolar model. In addition, observed cross-peaks (viii) and (ix) are from Ala H β to Gly H α 1 and H α 2 which are longer than 4 Å in the polar model, but about 2.3 and 2.6 Å, respectively, in the antipolar model. Thus, the 1H–1H distance information obtained from DQMAS 1H NMR clearly demonstrates that Silk II is antipolar, not polar. Thus, this is different from the Silk II model proposed by authors in this paper.

Fig.

Lotz supported the antipolar stems and lamellar model proposed by Asakura et al. as follows (ref. Lotz.B, ChemBloChem, 2022, e202100658).

Using the Marsh et al. model and taking the experimental total distance of 8.96 Å, minimization of the packing energy partitions this distance in 3.83 Å and 5.13 Å for the Gly-Gly and Ala-Ala intersheet distances, respectively. The matter is however more complex, since early on a possible structural disorder was considered. A fraction of the stems are in antipolar orientation, with their alanine residues located on the “glycine” side. Indeed, later X-ray analyses led to a reevaluation of the Marsh et al. model and conclude that the sheets feature a systematic antipolar antiparallel stem orientation. In addition, ¹³C solid-state NMR spectroscopy helps reach the detailed “heterogeneous local structure along the chain”, since it differentiates the packing of polar and

antipolar stems, and establishes the presence of distorted β turns in the chain folds: modern investigation techniques have reached a truly intimate analysis of the silk crystal structures.

The review containing Tables and Figures was attached as file.in Review attachment (optional).

Reviewer #3 (Remarks to the Author):

Review of: Silk Road Revealed: Mechanism of silk fibre formation in *Bombyx mori*.

General remarks:

The authors have conducted a comprehensive study on the structure and behaviour of new protein-native-like silk fibroin and compared its characteristics to regenerated silk fibroin. This is a very elegant and convincing study. Congratulations! My major remarks are as follow:

1) The authors convinced me that NLSF has better structural and behavioural characteristics than RSF obtained via the standard protocol. However, they do not provide any information, besides the washing step, on how NLSF has been obtained. Without this information, how can readers understand what is the origin of the reported differences?

2) Through the entire text, the authors claim that the NLSF assembly and characteristics are similar or close to those of NSF. No data have been provided for NSF. If the authors do such a comparison based on previously published work, they should at least show published graphs (with annotation adopted from...) together with NLSF graphs/structures in the SI, as they did in Figure 5.

3) The authors show TEM images of diluted NLSF assembled into “solenoid” structures. No images of diluted RSF assembled under the same conditions have been provided. Also, a similar structural organization has been previously reported for NSF (<https://doi.org/10.1016/j.scib.2023.12.050>). I am wondering what the authors’ opinions are regarding the possible alignment of these assemblies during the spinning process (parallel or perpendicular to a microfiber axis)?

4) The title does not reflect the content of the manuscript. The study describes the structural and behavioural characteristics of new type of protein, which is native-like silk fibroin.

To sum up, the authors claim, via multiple analysis, that NLSF is better than RSF and behaves similarly to NSF. Without providing 1) information on how NLSF has been obtained it’s difficult to understand what makes NLSF better than RSF; 2) the authors compare NLSF and the available literature on NSF without providing the possibility of a direct comparison (graphs against graphs, structure against structure). The latter makes reading the manuscript extremely difficult, jumping from one paper to another.

Minor remarks:

5) Line 22: a novel β -solenoid structure. I suggest to rephrase this statement, specifically “novel”. β -solenoid fold has already been studied and reported for amyloids.

6) Line 23-24: It is unclear what promotes what? The fact of the presence of the N-terminal or acidification or its hydrolysis upon acidification? This statement should be clarified.

- 7) Line 39: The authors mention the role of pH and metal ions in silk fibre formation, but cite only 1 work. I suggest to expand the citation list.
- 8) Line 62: What do "(14,15)" refer to?
- 9) Line 63: It appears in the track changes mode.
- 10) Line 69: It's unclear what the authors mean by "optical texture". If they mean pattern, then what kind of pattern or is it an ensemble of patterns?
- 11) Line 70-72: The authors claim that silk undergoes transformation from the "isotropic phase" to an optical texture when it flows from the posterior through the middle towards the anterior section of the silk gland. I am not entirely sure that in the posterior and middle sections there is an isotropy in silk. It has not been experimentally proven nor disproven. Maybe the authors mean that patterns appear only in the anterior section.
- 12) Line 85: What do the authors mean by "cellular texture"?
- 13) The authors discuss the hydrolysis of the fibroin during degumming of the cocoon fibres in the presence of Na_2CO_3 and refer to elimination of the hydrolysis event in NLSF preparation. Why? In the experimental section they provide detailed information on how RSF has been prepared (degumming, regeneration, and washing steps) but NO information on the preparation protocol for NLSF. Without the experimental details, it is unclear why and how hydrolysis has been eliminated.
- 14) What samples are shown in Figure S23? If the images depict highly diluted samples of NLSF, the question is how is RSF organised under the same conditions, namely, at the similar concentrations, pH, and buffer (if any).
- 15) Figure 1. Do the shown structural data refer to NSF, RSF, or NLSF?
- 16) Figure 3 (A) line 226: It is unclear what sample has been analysed? RSF or NLSF?
- 17) Generally, the authors should revise the legends for the figures in the main text and in the SI. It's unclear what sample has been analysed.

Review document, answers to Reviewer #1

Referee #1: Expert in mechanics of spider silk

Reviewer #1 (Remarks to the Author):

The manuscript “Silk road revealed: mechanism of silk fibre formation in *Bombyx mori*” by Moreno-Tortolero et al. may be acceptable for publication in *Communications Biology* after minor revision.

The authors present a comprehensive study on the rheology of fibroin solutions, as well as on the microstructures obtained from these solutions. The number of characterization techniques is extensive and the results are convincing, so that some aspects of the spinning process of *Bombyx mori* silk seem to be clarified from this work. In this regard, I have just a couple of suggestions to enhance the readability of the manuscript.

Dear reviewer, we very much appreciate your expertise and the time you have taken to review our manuscript, as we do the positive feedback. We believe the suggested changes will make our manuscript stronger and we hope to have addressed them to your satisfaction.

Reviewer 1, point 1:

The title seems slightly misleading in its present form. I would suggest a more informative title, such as “Molecular organization of fibroin heavy chain and the mechanism of silk fibre formation in *Bombyx mori*” or similar.

Answer:

We have now changed the title to reflect the content as per your suggestion. Many thanks for the example. New title is: Molecular organization of Silk-I fibroin heavy chain and mechanism of fibre formation in *Bombyx mori*.

Reviewer 1, point 2:

Page 5, Line 172. In addition to the references mentioned after the sentence “It has largely been recognised that in native spinning both shear stress and pH changes are responsible for the transformation of the soluble fibroin into the solid fibre”, and due to the central role of this hypothesis on the spinning of *B. mori* silk, the following reference is relevant: Cenis et al. *Soft Matter* 11 (2015) 8981. This work is a clear evidence that it is possible to obtain a silk fiber by simply immersing the silk gland in a mild acidic solution and stretching it. Consequently, no additional requirement is necessary for the fluid to solid transition during spinning.

Answer:

We appreciate the attention to detail offered here and we agree that this reference does indeed offer clear evidence that acidification and stretching are enough. We have added this reference now as suggested.

Line 169: It has largely been recognised that in native spinning both flow stress and pH changes are responsible for the transformation of the soluble fibroin into the solid fibre.^{7,52–54}

Reviewer 1, point 3:

Page 8, Figure 4. This Figure is critical to all the work and, consequently, it should be absolutely clear to the reader. In this sense, Figure 4 should reflect the model proposed by the authors with respect to the supramolecular organization of the proteins. In its present form, it is not sufficiently clear the transition from the single molecule to the tetramer and especially, between the tetramer and the multimer. In particular, the geometry of individual proteins should be clearly shown in the multimer, including the situation of selected N-terminal domains and some solenoids.

Answer:

Thank you for highlighting this, we agree that these molecular structures were challenging to see the individual components and as a result we have changed substantially Figure 4 and 5 to aid in the visualisation of the molecular transitions, from the single molecule to the multimer.

Figure 1. Evidence of a reversible pH-driven assembly in NLSF. (A) Cartoon of Fibroin heavy Chain with 12 repetitive blocks (blue), 11 flexible linkers (red), NTD (green) and CTD (purple), under this a molecular cartoon of one of the folded models corresponding to NTD and first repetitive domain in a beta-solenoid

type of fold. (B) Negative stained TEM of NLSF at pH 8. (C) Negative stained TEM of NLSF at pH 6. (D) Surface representation of the proposed biological unit of the NTD tetramer⁶² coloured by hydrophobicity (cyan hydrophytic and wheat hydrophobic) with top dimer represented as a cartoon (left) and as seen from the back (right) with white arrows pointing at hydrophobic patches involved in dimer-dimer (left) and tetramer-tetramer (right). (E) Proposed NTD tetramer stacking shows 2 tetrameric units stacked from the side (left) and the top (right) with arrows indicating the twist of the stacking. (F) Negative stained TEM of NLSF at pH 6 showing supramolecular brush-like structures formed via NTD stacking. (G) Molecular model of the transition, left to right, from monomer (n1), dimer (n2), tetramer (n4) and octamer (n8) coloured by chain (n1-4), and by tetrameric unit (n8) and a cross-section of n8 indicating the dimer-dimer (n2/n2) and tetramer-tetramer (n4/n4) interfaces by discontinuous arrows and the stacking direction by continuous arrows on the far right; n4 and n8 models also show a ghostly representation of their surface. (H) Proposed pH driven self-assembly of the protein, from globular unimer, extended conformation, dimer (each molecule donates two β -hairpins that interlock), tetramers (as shown in D, formed by dimer units stacking with a central symmetry plane), octamer (as shown in E, formed by the stacking of tetrameric units), and multimeric bottlebrush-like fibril showing proposed cholesteric order (left to right). Scale bars 100 nm for B and C, and 500 nm for F.

And

Figure 2. Summary of the newly proposed assembly pathway for fibroin into the silk fibre. Cartoon with the schematic representation of the proposed self-assembly pathway for fibroin. In the first instance, fibroin at pH greater than 7 exists as a globular unit. At pH lower than 7 extend and form supramolecular brush-like fibrillar structures that would align with the direction of the flow and originate a cholesteric texture. At critical stress, the supramolecular structures break, and the formed solenoid network aligns under flow to generate the nematic order. Illustration with optical textures was reprinted (adapted) with permission from Asakura et al. Some Observations on the Structure and Function of the Spinning apparatus in the Silkworm *Bombyx mori*, 2007 *Biomacromolecules*, ACS. Copyright 2007 American Chemical Society.²⁸

Additionally, I suggest the following minor amendments:

Reviewer 1, point 4:

Page 2, Line 60. Silk-1 -> Silk-I

Answer:

Change has been made

Reviewer 1, point 5:

Page 7, Line 227. Carton -> Cartoon

Answer:

Change has been made

Review document, answers to Reviewer #2

Referee #2: Expert in NMR structure of silk

Reviewer #2 (Remarks to the Author):

The review of this paper is as follows, and the reviewer completely disagree with the Silk I and Silk II structural models proposed in the paper for the following reasons.

In this paper, using the AlphaFold2 simulation of the repetitive domain of FibH, the authors found that these repetitive domains are predicted to adopt a β -solenoid conformation. They said that the predictions showed low confidence, and relatively high predicted Local Distance Difference Test (pLDDT) in average of about 50.

Namely, this is an initial prediction with low confidence, and further experimental verification is clearly needed. This result is an important step in understanding the structure and function of proteins because the proposal of the structural model should be done very carefully using many experimental data about Silk I.

However, their experimental data to determine Silk I structure in molecular level used by them is quite few, only simple X-ray diffraction data and solution NMR chemical shift data. Therefore, more reported experimental data should be used to check the model.

In the process to propose type II b-turn model for Silk I, Asakura et al. used many experimental data and therefore, the b-solenoid model should be checked by comparing with these experimental data.

Dear reviewer, we thank you for your expertise and the time you have dedicated to reviewing our work.

We truly appreciate your comments and perspective and, in this revision, hope to have addressed the perceived low confidence of the initial AlphaFold predictions through further experimental verification using TEM, solution NMR, fibre X-ray diffraction, and Circular dichroism.

We also deeply apologise the lack of clarity regarding the intensions of our work in the original version of our manuscript. We have amended this to be clearer now. We have not proposed a new model for this Silk-II polymorph and have now modified the title to "Molecular organization of Silk-I fibroin heavy chain and the mechanism of silk fibre formation in *Bombyx mori*."

We are also very familiar with the foundational and important work of Prof. Asakura and his group. However, we hope to add our data and interpretation to the scientific community which will enable our models and hypothesis to be discussed and tested against existing and future literature. We feel that this is a worthwhile endeavour as there are some areas where previously published works support our hypothesis.

- 1- As discussed within our manuscript, the type-II beta turn model, similar to the beta sheets, is a lamellar system, where its stability is contingent on the presence of many molecules. Such a system does not explain the presence of a fold in a diluted state, nor it does, crucially, explain the observation of fibrillar structures by us and others. In fact, our model is consistent not only with our presented data, but also with observations reported from AFM (<https://onlinelibrary.wiley.com/doi/10.1002/%28SICI%291099-0488%2820000601%2938%3A11%3C1436%3A%3AAID-POLB30%3E3.0.CO%3B2-8>), and perhaps more informatively, from SAXS (see <https://pubs.acs.org/doi/full/10.1021/ja806654t>

for *Bombyx mori* SAXS experiments, and interpretation of similar scattering patterns as worm-like micelles in <https://www.ncbi.nlm.nih.gov/pmc/articles/PMC7421538/>). Moreover, we note that a similar model has been proposed recently, independently, for GR peptides with repeat lengths of 20 and above using a combination of CD, SAXS, AlphaFold and Molecular dynamics (<https://www.science.org/doi/full/10.1126/sciadv.adj0347#sec-4>). Note in this work, in Figure S 2, the notable similarities of the presented SAXS data with that reported for native Silk fibroin in <https://pubs.acs.org/doi/full/10.1021/ja806654t>.

- 2- The type-II beta turn model proposed by Asakura et al. appears to us to differ from CD profiles. The observed CD spectra from silk fibroin, with a known minima around 195 nm is not commensurate with an extensive type-II beta turn fold, which would present a minima between 160 to 180 nm and a maxima between 180 and 200 nm, as presented in <https://pubs.rsc.org/en/content/articlelanding/2020/cp/c9cp05776e>. The CD spectra is, however, consistent with a superfold of polyproline-II or poly-glycine-II-like strands, as we propose in this work, and similarly interpreted recently for a similar system (polyGR work above).

Reviewer 2, point 1:

Asakura et al. used many REDOR experimental data. Many ^{13}C ^{15}N distance informations were obtained experimentally and calculated for type II b-turn model. The agreement between the observed and calculated distances was excellent as summarized in Table 1 (T.Asakura et al. *Macromolecules*, 2005, 38, 7397-7403).

Table 1 and Fig.

T. Gullion et al. (Gullion and Asakura et al. *JACS*, 2003, 125, 7510-7511) also tried ^{13}C REDOR experiment and supported type II b-turn structure of Silk I.

The calculated ^{13}C .. ^{15}N distances for b-solenoid model using the torsion angles reported in Fig.S10 should check the agreement with the observed many distance data.

Answer:

We appreciate the contributions noted by the reviewer in this point. We understand that in these works, the authors used peptide models of relatively limited size, namely, GA15, and although we believe these might serve as models for Silk-I, there is a gap in the evidence that the length of the peptides recapitulates, truly, the conformation of silk fibroin considering recent evidence where the fold might be stabilised after a critical length (<https://www.science.org/doi/full/10.1126/sciadv.adj0347#supplementary-materials>). For this reason, we refrained from referencing structural work derived from the use of peptides. Moreover, we note that, like NOESY in the liquid state, the assignment of distances to particular residue positions on such a repetitive system becomes difficult, as it is impossible to distinguish between intra-strand (assumption made in the referenced works by reviewer) or across-strand distances, i.e. it is possible to justify several possible packings using these distances. See our answer to point 4 as well.

Reviewer 2, point 2:

The type II β -turn model can explain the observed (Iobs) and calculated (Icalc) structure amplitudes using X-ray diffraction data of Silk I film very well (T.Asakura et al. *Macromolecules*, 2005, 38, 7397-7403).

Table

The authors should try these checks using their X-ray diffraction data of Silk I.

Answer:

We thank the reviewer for their suggestions for validating our interpretation. Several models have been proposed that account for some of the observed reflections in X-ray and electron diffraction and we have now added a few lines to better portray the breadth of interpretation for this type of data within the literature.

lines 63-69, and Table S1.

“Over time, several models have been proposed to account for the diffraction patterns (summarised in Table S1), however, all of these suffer from limitations inherent to the technical difficulties in acquiring aligned fibre X-ray diffraction patterns or complete datasets from electron diffraction. Most of the proposed unit cells so far belong to low-symmetry systems, perhaps due to the initial required assumptions necessary to arrive to structural model given the limited data available. Currently, the most accepted Silk I model is a type-II β -turn rich structure,²⁹ similar to the crankshaft model proposed by other before.^{30,31”}

Our model accounts for all the observed reflections, and we have added a table with the predicted reflections using our proposed unit cell and their corresponding index, showing that they overlap with experimental fibre X-ray diffraction pattern (Table S2, see below). We note that our proposed unit cell is similar to the almost extended strand proposed by Guido et al, although their data (electron diffraction) suggested a unit cell containing 6 stacked strands, as well as the unit cell proposed before for polyGlycine-I polymorph, as discussed within our manuscript.

Table S2

Reviewer 2, point 3:

The another experimental data are ^1H DQMAS NMR data of Silk I although they did not cite in their paper. (T. Asakura et al., Determination of Accurate ^1H Positions of (Ala-Gly) $_n$ as a Sequential Peptide Model of Bombyx Mori Silk Fibroin before Spinning (Silk I). *Macromolecules* 2013, 46, 8046–8050). In the paper, Asakura et al. described as follows. The accurate ^1H positions of alanine-glycine alternating copolyptide, (AG) $_{15}$ with Silk I structure were determined. For the purpose, the geometry optimization was performed starting with the atomic coordinates of the hetero atoms reported previously (T.Asakura et al. *Macromolecules* 2005, 38, 7397–7403, *J. Mol. Biol.* 2001, 306, 291-305) and applied only for protons under periodic boundary conditions. The agreement between the calculated and observed chemical shifts of all ^1H , ^{13}C and ^{15}N nuclei was excellent, indicating strongly that the determination of all the atomic-coordinate including ^1H nuclei was performed with high accuracy. Here the ^1H chemical shift was obtained by using both 1 mm microcoil MAS NMR probe-head for mass-limited solid-state samples developed by Asakura et al. and ultrahigh field NMR at 920 MHz. The DQ correlations in the ^1H DQMAS NMR spectra were also used to confirm the intra and intermolecular structures obtained here. The authors should check the b-solenoid model using these experimental data.

Answer:

We would kindly point the reviewer to our answer to point 1 regarding our choice to draw primarily on comparison to studies that have used silk proteins as opposed to peptide models.

Reviewer 2, point 3:

The solution NMR results of Silk I (Suzuki and Asakura et al. *Biomacromolecules*, 2014, 15, 104-112) also proposed type II b-turn structure. This was obtained from the observed chemical shifts ($^{13}\text{C}_\alpha$, $^{13}\text{C}_\beta$, $^{13}\text{C}_\text{O}$, $^1\text{H}_\alpha$, $^1\text{H}_\text{N}$ and ^{15}N) of the backbone atoms using TALOS-N. The torsion angle of Gly residues (77° , 10°) was obtained from solution NMR chemical shifts as well as the 2D diffusion NMR and REDOR experiments of solid-state NMR with very careful manner.

Answer:

We appreciate the mention of this reference, as we have tremendous respect for it, as it really helped us with the analysis of the solution state NMR data. Our assigned chemical shifts are consistent with the assignment made there, and similarly, we performed the TALOS-N predictions too, to arrive to the same predictions, which is the reason we didn't include the data.

Below we have extracted from the reference the figure containing the torsional angles predicted from TALOS-N software, where Suzuki and Asakura et al. circled in red the angles that correspond to the type-II beta turn and we have pointed with blue arrows the second population of torsional angles found for Glycine residues, which are commensurate with our model. Referenced work (<https://pubs.acs.org/doi/10.1021/jp0125395>) shows DFT calculations resulting in determination that torsional angles obtained for type-II beta turn are unlikely to represent the real structure, they, however, arrived to dihedral angles and a structure that is consistent with ours.

Asakura et al. dihedral angle predictions from TALOS-N

This works dihedral angles

We further note that in this same work, NOEs were used to strengthen the argument for the type-II β -turn model; we have extracted the NOESY Figure a below directly from the referenced work. The argument was based on the observation of NOEs between Alanine $H\alpha/HN$ with Serine, Tyrosine and Valine respective $H\alpha/HN$. The interpretation of these NOEs was based on the assignment of these to i and $i+2$ positions within the same strand.

However, we observed the same NOEs and we have interpreted these in a different way. Based on our model, the unit cell corresponds to an almost extended polyglycine-II-like strand forming a superhelix, within the proposed helix/solenoid, each strand is stacked on top of each other, with three of these strands forming the rungs of the helix. This assembly leaves each hexapeptide GAGXGA aligned in such a way that each X position is close to the previous, and thus, NOEs are observable between these position; i.e. position i to position $i+12$, as shown in the figure below. Thus, as we discussed in point 1, the correct assignment of nuclei distances to sequence in such a repetitive system can be ambiguous and warrants bringing into the general discussion for the field.

About Silk II model, T. Asakura et al. reported recently from the following references that Silk II is a lamellar structure.

We apologise for any confusion here. We have absolutely not proposed a new model for the structure of Silk-II. We have no disagreement with the proposed model by Asakura et al. on regards to this polymorph and thus we have removed the molecular model showing a polar antiparallel structure in Figure S28 and again apologise for any confusion surrounding the intent of this work.

1. Asakura, T.; Aoki, A.; Komatsu, K.; Ito, C.; Suzuki, I.; Naito, A.; Kaji, H. Lamellar Structure in Alanine–Glycine Copolypeptides Studied by Solid-State NMR Spectroscopy: A Model for the Crystalline Domain of Bombyx Mori Silk Fibroin in Silk II Form. *Biomacromolecules* 2020, 21, 3102–3111.
2. Asakura, T.; Ogawa, T.; Naito, A.; Williamson, M. P. Chain-Folded Lamellar Structure and Dynamics of the Crystalline Fraction of Bombyx Mori Silk Fibroin and of (Ala-Gly-Ser-Gly-Ala-Gly)_n Model Peptides. *Int. J. Biol. Macromol.* 2020, 164, 3974–3983.
3. Asakura, T.; Williamson, M. P. A review on the structure of Bombyx mori silk fibroin fiber studied using solid-state NMR: An antipolar lamella with an 8-residue repeat, *Int. J. Biol. Macromol.* 2023, 245, 125537.
4. Suzuki, Y.; Morie, S.; Okamura, H.; Asakura, T.; Naito, A. Real time monitoring of the structural transition of Bombyx mori liquid silk under pressure by solid-state NMR, *J. Am. Chem. Soc.* 2023, 145, 22925-22933.)

In addition, the 1H–1H distance information obtained from DQMAS 1H NMR clearly demonstrates that Silk II is antipolar, not polar as follows. This is different from the Silk II model (polar stem) proposed by authors.

In order to distinguish between these models, antipolar or polar, Asakura et al. obtained 1H–1H distance information from double-quantum magic angle spinning (DQMAS) 1H NMR spectra. In these spectra, only 1H–1H distances within about 4 Å give rise to observable cross-peaks. A set of nine 1H–1H correlation signals from the 1H DQMAS NMR spectrum of (AG)₁₅ is marked in Fig. 6 (Ia). These signals were compared to the distances calculated from the polar Marsh and antipolar Takahashi models. Fig. 6(II) shows that the Gly H α 2 protons in adjacent β -strands are very close to one another (about 2 Å) in the polar Marsh model and therefore a diagonal peak for Gly H α 2 should be detected. However, in the anti-polar model this distance is more than 4 Å, and therefore no peak is expected. This feature is difficult to distinguish from Fig. 6(Ia) because the Gly H α 2 and Ala H α peaks have overlapping chemical shifts. Asakura et al. therefore synthesized deuterium-labeled ([2-d]AG)₁₅ and acquired another DQMAS 1H NMR spectrum (Fig. 6(Ib)). There is clearly no Gly H α 2 peak on the diagonal, now that the Ala H α signal at around 5.0 ppm has been removed. This observation provides very strong evidence that the polar model cannot be correct. Other experimental 1H–1H distance data were also only compatible with the antipolar model. For example, the observed cross-peak (v) is between Gly H α 2 and Ala H α which is longer than 4 Å in the polar model, but about 2.4 Å in the antipolar model. In addition, observed cross-peaks (viii) and (ix) are from Ala H β to Gly H α 1 and H α 2 which are longer than 4 Å in the polar model, but about 2.3 and 2.6 Å, respectively, in the antipolar model. Thus, the 1H–1H distance

information obtained from DQMAS ^1H NMR clearly demonstrates that Silk II is antipolar, not polar.
Thus, this is different from the Silk II model proposed by authors in this paper.

Fig.

Lotz supported the antipolar stems and lamellar model proposed by Asakura et al. as follows (ref. Lotz.B, ChemBioChem, 2022, e202100658).

Using the Marsh et al. model and taking the experimental total distance of 8.96 Å, minimization of the packing energy partitions this distance in 3.83 Å and 5.13 Å for the Gly-Gly and Ala-Ala intersheet distances, respectively. The matter is however more complex, since early on a possible structural disorder was considered. A fraction of the stems are in antipolar orientation, with their alanine residues located on the “glycine” side. Indeed, later X-ray analyses led to a reevaluation of the Marsh et al. model and conclude that the sheets feature a systematic antipolar antiparallel stem orientation. In addition, ^{13}C solid-state NMR spectroscopy helps reach the detailed “heterogeneous local structure along the chain”, since it differentiates the packing of polar and antipolar stems, and establishes the presence of distorted β turns in the chain folds: modern investigation techniques have reached a truly intimate analysis of the silk crystal structures.

The review containing Tables and Figures was attached as file.in Review attachment (optional).

Finally, we thank the reviewer again for their time and insights kindly given for our work. It is really appreciated and has helped us question, discuss and develop our thoughts and ideas surrounding our data and hopefully helped clarify our interpretation.

Review document, answers to Reviewer #3

Referee #3: Expert in self-assembly of biological materials

Reviewer #3 (Remarks to the Author):

Review of: Silk Road Revealed: Mechanism of silk fibre formation in *Bombyx mori*.

General remarks:

The authors have conducted a comprehensive study on the structure and behaviour of new protein-native-like silk fibroin and compared its characteristics to regenerated silk fibroin. This is a very elegant and convincing study. Congratulations!

Dear reviewer, we appreciate your expertise and the time you have taken to review our manuscript. Thank you for your kind words.

My major remarks are as follow:

Reviewer 3, point 1:

The authors convinced me that NLSF has better structural and behavioural characteristics than RSF obtained via the standard protocol. However, they do not provide any information, besides the washing step, on how NLSF has been obtained. Without this information, how can readers understand what is the origin of the reported differences?

Answer:

We appreciate this feedback, as it does indeed show a keen eye on detail. We acknowledge that reviewer is correct in their observations. The particular material we are using is under intellectual property protection and is covered by patent number US8,128,984B2. We direct the reviewer to claims 3 and, in particular, claim 7 of the patent. To obtain native-like behaviour it is essential to minimize hydrolysis of FibH, which is known to happen predominantly during degumming, and this can be achieved via the use of enzymes, milder degumming agents, or shorter degumming times.

To reflect this, we have modified the Materials section regarding NLSF by adding the following statement:

Briefly, silk fibres were initially treated as per patent number US 8,128,984B2 using mild degumming conditions to minimize fibroin hydrolysis, after washing with MilliQ water, fibres were gently dried and dissolved in 8.5 M LiBr at 60 °C over 4 hours.

Reviewer 3, point 2:

Through the entire text, the authors claim that the NLSF assembly and characteristics are similar or close to those of NSF. No data have been provided for NSF. If the authors do such a comparison based on previously published work, they should at least show published graphs (with annotation adopted from...) together with NLSF graphs/structures in the SI, as they did in Figure 5.

Answer:

The reviewer makes a perfectly valid point and highlights a challenge for the community. Currently only limited data exist for native silk extracted from the posterior section of the middle gland. However reassuringly, under these conditions, the protein is found at pH close

to neutral and viscoelastic liquids are seen, like our pH 7 liquid state. We are actively developing tools to test NSF response to measured environmental changes (Holland has 2 PhD students on it at the moment) but we feel these significant technical challenges lay beyond the scope of this study.

Reviewer 3, point 3:

3) The authors show TEM images of diluted NLSF assembled into “solenoid” structures. No images of diluted RSF assembled under the same conditions have been provided.

Answer:

Thank you for raising this, we have not included RSF images as previously published works have shown, for RSF, only small, featureless blobs are observed, which have given rise to the micellar model. Examples of literature where RSF has been studied using TEM, and the effect of pH has been observed: <https://link.springer.com/article/10.1007/BF03218556>, <https://link.springer.com/article/10.1007/s10895-021-02841-x#Sec12>.

We have added this comment and reference to the manuscript.

Line 291:

In contrast, RSF is known to be insensitive in this pH range, only showing featureless globular morphologies,^{82,83} consistent with the hypothesis that the protein is losing important structurally functional domains.

Also, a similar structural organization has been previously reported for NSF (<https://doi.org/10.1016/j.scib.2023.12.050>). I am wondering what the authors’ opinions are regarding the possible alignment of these assemblies during the spinning process (parallel or perpendicular to a microfiber axis)?

Great observation, we truly appreciate the interest of the reviewer! We were delighted with this work as it confirms the native-like behaviour of our material and been closely following from the preprint (<https://www.biorxiv.org/content/10.1101/2021.03.08.434386v1.full>). In the preprint, the authors suggest that the direction of the main axis of the multimer is coaligned with the flow, and we believe this orientation to be the most likely, but this differs in the published version. At the time we asked the senior author, Prof. He Huawei their thoughts about the alignment of the assemblies and thought it useful for the reviewer to share here under the confidentiality of peer review. An excerpt of our communication is presented below:

Prof. Huawei’s answer: “Indeed, any orientation of the "herringbone" structure is possible in both the preprint and the final version (Fig. 1), and currently we have no conclusive evidence to determine its precise orientation. We believe that the formation of the "herringbone" structure is due to certain factors during the flow process of NSF nanofibrils, in which the long axis of the NSF nanofibrils tends to be parallel to the direction of the spinning duct (similar to the trees drifting in a river, whose long axes are usually aligned with the direction of the water flow). As a result, the main axis of the formed "herringbone" structure is perpendicular to the direction of NSF nanofibrils flow. Therefore, we believe that the schematic in the final version may represent the real state of NSF nanofibrils in the silk gland lumen. In the future, we hope to experimentally determine the precise orientation of the "herringbone" structure”

Although we haven't communicated again with Prof. Huawei, we have tremendous appreciation for his efforts in answering our inquiry on such an honest way.

Addressing the reviewer comment specifically, we believe that under flow, such long supramolecular fibres would be under substantial torsional stresses if these were oriented perpendicular to the flow, given the shear stress distribution along the diameter of a tube. The resulting torque would align the main axis (longest dimension) of the structure in the direction of the flow. Following Prof. Huawei's analogy of a tree flowing down a river, it is the trunk of the tree and not the branches the ones that align with the flow.

However, as a result of the novelty and timeliness of their work, we are opting not to discuss this topic further within the manuscript as not to foreshadow and reduce any future impact from their group, especially when they have been so collegiate. Thus we prefer to keep this discussion and our communications with Prof. Huawei within the confidential confines of this peer review.

Reviewer 3, point 4:

The title does not reflect the content of the manuscript. The study describes the structural and behavioural characteristics of new type of protein, which is native-like silk fibroin.

Answer:

Similar to reviewer 1 comment, we have now changed the title to "Molecular organization of Silk-I fibroin heavy chain and mechanism of fibre formation in *Bombyx mori*" as suggested (see reviewer 1 remarks to authors, point 1). Regarding the suggestion that we should emphasise that this is for NLSF, we believe that we have demonstrated that through careful processing, retention of the native structure is maintained. Our material behaves in the same manner as a native, pre-spun, silk solution at different length-scales; from the observations of similar structures in TEM as the reviewer points to above, to the observations of very similar macroscopic behaviour (rheology).

To sum up, the authors claim, via multiple analysis, that NLSF is better than RSF and behaves similarly to NSF. Without providing 1) information on how NLSF has been obtained it's difficult to understand what makes NLSF better than RSF; 2) the authors compare NLSF and the available literature on NSF without providing the possibility of a direct comparison (graphs against graphs, structure against structure). The latter makes reading the manuscript extremely difficult, jumping from one paper to another.

We hope to have addressed the reviewers concerns above with respect to the production of our NLSF and although we definitely agree that a direct comparison to NSF would be useful, due to the technical challenges of NSF work such an endeavour is beyond the scope of this study.

Minor remarks:

Reviewer 3, point 5:

Line 22: a novel β -solenoid structure. I suggest to rephrase this statement, specifically “novel”. β -solenoid fold has already been studied and reported for amyloids.

Answer:

We apologise for the confusion, in the field of silk we believe this proposed structure is novel.

We have amended the text accordingly: Line 23

Our results have led us towards the novel proposition that the Silk-I fibroin heavy chain (FibH) from the silkworm, *Bombyx mori*, folds into a β -solenoid structure, where the N-terminal domain (NTD) templates reversible higher-order oligomerization driven by pH reduction.

Reviewer 3, point 6:

Line 23-24: It is unclear what promotes what? The fact of the presence of the N-terminal or acidification or its hydrolysis upon acidification? This statement should be clarified.

Answer:

We apologise for the lack of clarity in this instance. We have replaced the word promotes with templates in the abstract: Line 25

Our results have led us towards the novel proposition that the Silk-I fibroin heavy chain (FibH) from the silkworm, *Bombyx mori*, folds into a β -solenoid structure, where the N-terminal domain (NTD) templates reversible higher-order oligomerization driven by pH reduction.

Reviewer 3, point 7:

Line 39: The authors mention the role of pH and metal ions in silk fibre formation, but cite only 1 work. I suggest to expand the citation list.

Answer:

Thank you for the suggestion. We have increased the number of references here. Line 40

...but where tight control over pH and metal ion concentrations is exerted.¹⁻⁵

Reviewer 3, point 8:

Line 62: What do “(14,15)” refer to?

Answer:

Sorry, this was a formatting error on our behalf, has been updated now.

Reviewer 3, point 9:

Line 63: It appears in the track changes mode.

Answer:

Corrected

Reviewer 3, point 10:

Line 69: It's unclear what the authors mean by "optical texture". If they mean pattern, then what kind of pattern or is it an ensemble of patterns?

Answer:

Here we opted to use the word texture as opposed to pattern as a more common term for the liquid crystal field to describe gross spatial differences in molecular orientation. For clarity we have now amended the text as follows: Line 69

Moreover, the observation of a range of distinct liquid *crystalline* textures (patterns) *in vivo*

Reviewer 3, point 11:

Line 70-72: The authors claim that silk undergoes transformation from the "isotropic phase" to an optical texture when it flows from the posterior through the middle towards the anterior section of the silk gland. I am not entirely sure that in the posterior and middle sections there is an isotropy in silk. It has not been experimentally proven nor disproven. Maybe the authors mean that patterns appear only in the anterior section.

Answer:

We have removed the sentence: Silk goes from an isotropic texture in the posterior and middle gland sections to a series of complex transitioning optical textures. We have also clarified that the liquid crystalline textures have been observed for the anterior section of the gland.

The paragraph now reads:

Moreover, the observation of a range of distinct liquid crystalline textures *in vivo* at the start of anterior section of the gland and the spinneret remains unexplained,¹ with the mesogenic structures being unidentified. At the start of the anterior section of the silk gland, a "cellular optical texture" is observed, which transforms to an isotropic phase prior to reverting to a fully nematic phase before the spinneret². The emergence of the cellular optical texture has been attributed to epitaxial anchoring of rod-like mesogens under confinement³. However, this model does not adequately account for the subsequent transition to a nematic texture, under flow, as the tube diameter in the gland decreases towards the spinneret. Curiously, at a similar position to the cellular optical texture, evidence of cholesteric order has been observed using electron microscopy⁴. More importantly, despite the evidence of supramolecular order, a transition from the Silk I to Silk II polymorph only occurs later near the spinneret^{2,4}.

Reviewer 3, point 12:

Line 85: What do the authors mean by "cellular texture"?

Answer:

Cellular optical textures refer to specific patterns observed in some liquid crystalline systems that are distinctive from nematic phases. See reference by John Bunnig and John Lydon (<https://www.tandfonline.com/doi/abs/10.1080/02678299608032050>). This texture has been used to characterise the observed patterns that emerged after observing the carefully fixed gland content.

Reviewer 3, point 13:

The authors discuss the hydrolysis of the fibroin during degumming of the cocoon fibres in the presence of Na₂CO₃ and refer to elimination of the hydrolysis event in NLSF preparation. Why? In the experimental section they provide detailed information on how RSF has been prepared (degumming, regeneration, and washing steps) but NO information on the preparation protocol for NLSF. Without the experimental details, it is unclear why and how hydrolysis has been eliminated.

Answer:

Already discussed in point 1.

Reviewer 3, point 14:

What samples are shown in Figure S23? If the images depict highly diluted samples of NLSF, the question is how is RSF organised under the same conditions, namely, at the similar concentrations, pH, and buffer (if any).

Answer:

Figure S23 corresponds to NLSF. The caption now reflects this, and it reads: "TEM analysis of supramolecular assemblies observed in NLSF."

Reviewer 3, point 15:

Figure 1. Do the shown structural data refer to NSF, RSF, or NLSF?

Answer:

The experimental data was obtained using NLSF and the caption now reflects this, reading:

"Figure 3. Structural model for fibroin heavy chain. (A) Experimental and simulated Silk-I diffraction pattern obtained from drop-casted film of NLSF and derived unit cell showing the proposed fibre axis..."

In regard to whether the structure refers to NSF, RSF or NLSF, given that very similar X-ray diffraction patterns have been obtained from all through time, the structural model should represent the three.

Reviewer 3, point 16:

Figure 3 (A) line 226: It is unclear what sample has been analysed? RSF or NLSF?

Answer:

In Figure 3 (A), the data corresponds to NLSF and the caption has now been updated to reflect this. Caption now reads: "(A) Master curves created using oscillatory shear data of the liquid and solid-like samples of NLSF shown on the left and right, respectively."

Reviewer 3, point 17:

Generally, the authors should revise the legends for the figures in the main text and in the SI. It's unclear what sample has been analysed.

Answer:

All images that refer to experimental data on NLSF now contain the clarification within the caption.

Reviewers' comments:

Reviewer #1 (Remarks to the Author):

The manuscript is publishable in its present form

Reviewer #2 (Remarks to the Author):

The reviewer read the response to the reviewer's comments with great interest.

1. At first, it is necessary to define silk I and also silk I*.

Silk I was defined B. mori silk fibroin (SF) structure after drying of the liquid silk stored in middle silk gland in the solid state. In addition, silk I* is a newly defined structure of only the repeated sequences, (AGSGAG)_n in the solid state, which is almost half of the whole SF. The structure of silk I* is different from more heterogeneous silk I structure defined as the solid-state structure of whole SF before spinning. Namely silk I consist of silk I* plus other conformations, mainly random coil conformation in the solid state. Please refer the following papers..

Asakura, T.; Endo, M.; Hirayama, M.; Arai, H.; Aoki, A.; Tasei, Y. Glycerin-Induced Conformational Changes in Bombyx Mori Silk Fibroin Film Monitored by ¹³C CP/MAS NMR and ¹H DQMAS NMR. *Int. J. Mol. Sci.* 2016, 17, 1517–1533.

Nishimura, A.; Matsuda, H.; Tasei, Y.; Asakura, T. Effect of Water on the Structure and Dynamics of Regenerated [³⁻¹³C] Ser, [³⁻¹³C] , and [³⁻¹³C] Ala-Bombyx Mori Silk Fibroin Studied with ¹³C Solid-State Nuclear Magnetic Resonance. *Biomacromolecules* 2018, 19, 563–575.

Asakura, T.; Ogawa, T.; Naito, A.; Williamson, M. P. Chain-Folded Lamellar Structure and Dynamics of the Crystalline Fraction of Bombyx Mori Silk Fibroin and of (Ala-Gly-Ser-Gly-Ala-Gly)_n Model Peptides. *Int. J. Biol. Macromol.* 2020, 164, 3974–3983.

Asakura, T. Structure of Silk I (Bombyx Mori Silk Fibroin before Spinning) -Type II β-Turn, Not α-Helix-. *Molecules* 2021, 26, 3706.

Asakura, T.; Naito, A. Structure of silk I (Bombyx mori silk fibroin before spinning) in the dry and hydrated states studied using ¹³C solid-state NMR spectroscopy. *Int. J. Biol. Macromol.* 2022, 216, 282–290.

Anyway, the title should be changed to "Molecular organization of fibroin heavy chain and the mechanism of silk fibre formation in Bombyx mori" as reviewer 1 recommended.

2. Asakura et al. already checked (AG)₁₅ carefully whether or not it is suitable to the model of native SF with silk I* form.

The results are as follows.

c. Asakura et al. *Macromolecules*, 2013, 46, 8046

Thus, these points should be included in the revised paper.

Although there are other many points to discuss, the reviewer do not want to delay the publication of this paper. The reviewer agree with the author's opinion (we hope to add our data and interpretation to the scientific community which will enable our models and hypothesis to be discussed and tested against existing and future literature. We feel that this is a worthwhile endeavour as there are some areas where previously published works support our hypothesis.)

Reviewer #3 (Remarks to the Author):

Authors have fully addressed my comments. I recommend to accept.

Review document, answers to Reviewer #1

Reviewers' comments:

Reviewer #1 (Remarks to the Author):

The manuscript is publishable in its present form

Dear Reviewer, we are tremendously grateful for your time and contributions in reviewing our manuscript, and we believe we have a stronger manuscript after your review.

Review document, answers to Reviewer #2

Reviewer #2 (Remarks to the Author):

The reviewer read the response to the reviewer's comments with great interest.

Dear Reviewer, it is with great interest that we have also received the reviewer's comments, and we believe the reviewer's invaluable input has contributed to making our manuscript stronger.

Reviewer 2, point 1:

At first, it is necessary to define silk I and also silk I*.

Silk I was defined *B. mori* silk fibroin (SF) structure after drying of the liquid silk stored in middle silk gland in the solid state. In addition, silk I* is a newly defined structure of only the repeated sequences, (AGSGAG)_n in the solid state, which is almost half of the whole SF. The structure of silk I* is different from more heterogeneous silk I structure defined as the solid-state structure of whole SF before spinning. Namely silk I consist of silk I* plus other conformations, mainly random coil conformation in the solid state.

Please refer the following papers..

Asakura, T.; Endo, M.; Hirayama, M.; Arai, H.; Aoki, A.; Tasei, Y. Glycerin-Induced

Conformational Changes in *Bombyx Mori* Silk Fibroin Film Monitored by ¹³C CP/MAS NMR and ¹H DQMAS NMR. *Int. J. Mol. Sci.* **2016**, *17*, 1517–1533.

Nishimura, A.; Matsuda, H.; Tasei, Y.; Asakura, T. Effect of Water on the Structure and Dynamics of Regenerated [3-¹³C] Ser, [3-¹³C] , and [3-¹³C] Ala-*Bombyx Mori* Silk

Fibroin Studied with ¹³C Solid-State Nuclear Magnetic Resonance. *Biomacromolecules*

2018, *19*, 563–575.

Asakura, T.; Ogawa, T.; Naito, A.; Williamson, M. P. Chain-Folded Lamellar Structure and Dynamics of the Crystalline Fraction of *Bombyx Mori* Silk Fibroin and of (Ala-Gly-Ser- Gly-Ala-Gly)_n Model Peptides. *Int. J. Biol. Macromol.* **2020**, *164*, 3974–3983.

Asakura, T. Structure of Silk I (*Bombyx Mori* Silk Fibroin before Spinning) -Type II β-Turn, Not α-Helix-. *Molecules* **2021**, *26*, 3706.

Asakura, T.; Naito, A. Structure of silk I (*Bombyx mori* silk fibroin before spinning) in the dry and hydrated states studied using ¹³C solid-state NMR spectroscopy. *Int. J. Biol.*

Macromol. **2022**, *216*, 282-290.

Anyway, the title should be changed to “Molecular organization of fibroin heavy chain and the mechanism of silk fibre formation in *Bombyx mori*” as reviewer 1 recommended.

Answer

We have now removed the word Silk I from the title and the new title, as suggested, is: Molecular organization of fibroin heavy chain and mechanism of fibre formation in *Bombyx mori*.

We appreciate the reviewer's input, and acknowledge the definition of Silk I* as the ordered structure within Silk I. We have added to the discussion the following (Line 136-139):

Significantly, within each repetitive domain, there are more regular and seemingly rigid

subdomains (as shown in Figure S3) that might be the ones giving rise to the deconvolved solid-state NMR Silk I* structures, with the remainder of the dynamic and unordered subdomains contributing to the overall Silk I spectrum.^{52,53}

Reviewer 2, point 2:

Asakura et al. already checked (AG)₁₅ carefully whether or not it is suitable to the model of native SF with silk I* form.

The results are as follows.

- a. Asakura et al. *Macromolecules*, **2005**, *38*, 7397.

The IR and ¹³C CP/MAS NMR spectra of (AG)₁₅ and SF with silk I forms are shown. These data indicate clearly that (AG)₁₅ is the suitable model to describe the structure of SF with silk I form.

- b. Asakura et al. *Biopolymers*, **2001**, *58*, 521.

The ¹³C CP/MAS NMR spectra of A. (AG)₁₅ and B. Cp fraction with silk I* forms are shown. Here the Cp fraction is the crystalline fraction obtained after cleavage by chymotrypsin of whole SF and 56% of whole SF. These data indicate that (AG)₁₅ is the suitable model to describe the structure of Cp fraction with silk I* form.

- c. Asakura et al. *Macromolecules*, **2013**, *46*, 8046

The 2D ^1H (920 MHz) DQMAS NMR spectra of A. $(\text{AG})_{15}$ and B. Cp fraction with silk I* forms are shown. These data indicate that $(\text{AG})_{15}$ is the suitable model to describe the structure of Cp fraction with silk I form.

d. Asakura et al. *Biomacromolecules*, **2020**, *21*, 3102.

e. Asakura and Williamson, *Int. J. Biol. Macromol.* **2023**, *245*,125537.

Figure 1. Expanded Ala $C\beta$ peak in the ^{13}C solid-state NMR spectrum of the (a) SF fiber in the Silk II form, deconvoluted into two spectra, (b) and (c) of the crystalline (Cp) and noncrystalline fractions, respectively.^{23–26} The spectrum (b) of the Cp fraction (56% of the whole SF) splits into three peaks which consist of 32% distorted β -turn (t), 45% β -sheet (A), and 23% another β -sheet (B). The difference spectrum (c) obtained by subtracting the spectrum (b) from the whole SF spectrum (a) can be assigned to that of noncrystalline fractions.

Figure 2. ^{13}C CP/MAS NMR spectra of $(\text{AG})_{15}$, $(\text{AGSGAG})_5$, and the Cp fraction in the Silk II forms of SF together with their assignment.

Figure 3. Expanded Ala C β peaks in the ^{13}C CP/MAS NMR spectra of G(AG) $_2$ and (AG) $_m$ ($m = 6, 9, 12, 15,$ and 25) in Silk II forms. The deconvoluted Ala C β peaks of (AG) $_m$ ($m = 9, 12, 15,$ and 25) are shown as broken red lines.

The ^{13}C CP/MAS NMR spectra of SF (Figure 1), Cp fraction and (AG) $_{15}$ with silk II forms (Figures 2 and 3) are shown. The SF spectra are decomposed as the Cp fraction and noncrystalline fraction (Figure 1). These Figures show that the (AG) $_{15}$ with silk II form is the suitable model to describe the structures of SF and the Cp fraction with silk II forms.

Thus, these points should be included in the revised paper.

Answer

We understand that the similar conformation has been proven in the solid state, and to acknowledge this we have added the following statement in our conclusions (Line 416-417):

it has already been proven that (AG) $_{15}$ peptides show similar conformation to FibH in the solid state.^{52,54,103,104}

Nonetheless, we also note that the similarity of the conformation in the liquid-state remains to be explored. We also understand the technical challenge of characterising polyGA peptides in solution due to their insolubility and remarkable tendency to form structures (oligomers).

Although there are other many points to discuss, the reviewer do not want to delay the publication of this paper. The reviewer agree with the author's opinion (we hope to add our data and interpretation to the scientific community which will enable our models and hypothesis to be discussed and tested against existing and future literature. We feel that this is a worthwhile endeavour as there are some areas where previously published works support our hypothesis.).

Dear reviewer, we appreciate and respect very much your time, expertise, and your input through this review process, and we hope to have resolved, to your satisfaction, these last concerns.

Review document, answers to Reviewer #3

Reviewer #3 (Remarks to the Author):

Authors have fully addressed my comments. I recommend to accept.

Dear Reviewer, we are tremendously grateful for your time and contributions in reviewing our manuscript, especially in the detailed observations and interest offered through the review. We are convinced our manuscript is better after your input.